# Numerical Analysis of HiPPO-LegS ODE for Deep State Space Models

**Jaesung R. Park**                                             *ryanpark7@math.ucla.edu*
*Department of Mathematics*
*University of California, Los Angeles*

**Jaewook J. Suh**                                                  *jacksuh@rice.edu*
*Department of Computational Applied Mathematics & Operations Research*
*Rice University*

**Youngjoon Hong**                                              *hongyj@snu.ac.kr*
*Department of Mathematical Sciences*
*Seoul National University*

**Ernest K. Ryu**                                                 *eryu@math.ucla.edu*
*Department of Mathematics*
*University of California, Los Angeles*

**Reviewed on OpenReview:** *https://openreview.net/forum?id=83dhVASBPn*

## Abstract

In deep learning, the recently introduced state space models utilize HiPPO (High-order Polynomial Projection Operators) memory units to approximate continuous-time trajectories of input functions using ordinary differential equations (ODEs), and these techniques have shown empirical success in capturing long-range dependencies in long input sequences. However, the mathematical foundations of these ODEs, particularly the singular HiPPO-LegS (Legendre Scaled) ODE, and their corresponding numerical discretizations remain unsettled. In this work, we fill this gap by establishing that HiPPO-LegS ODE is well-posed despite its singularity, albeit without the freedom of arbitrary initial conditions. Further, we establish convergence of the associated numerical discretization schemes for Riemann integrable input functions.

## 1 Introduction

State-space representation is a cornerstone of dynamical-system theory and has been instrumental in the analysis and control of physical processes in control engineering, signal processing, and computational neuroscience. In the deep-learning literature, this classical framework has recently re-emerged as a promising paradigm for sequence modelling, offering a principled alternative to recurrent and attention-based architectures Gu et al. (2022); Dao & Gu (2024); Zhu et al. (2024); Nguyen et al. (2022); Goel et al. (2022). Modern state-space models for long sequences build on a synthesis of two pillars: (i) linear state-space theory in its canonical form Williams & Lawrence (2007); Zak et al. (2003) and (ii) the HiPPO (High-order Polynomial Projection) framework Gu et al. (2020), which prescribes optimal polynomial projections for compressing the history of an input signal. This amalgamation provides both an interpretable memory mechanism and a mathematically tractable route for capturing long-range dependencies within deep architectures.

HiPPO is a framework using an $N$-dimensional ordinary differential equation (ODE) to approximate the continuous-time history of an input function $f$. In particular, the HiPPO-LegS (Legendre Scaled) ODE is

$$c'(t) = -\frac{1}{t}Ac(t) + \frac{1}{t}Bf(t), \tag{1}$$

Table 1: Summary of convergence results for various discretization methods under different regularity assumptions on the input $f$. Rates are stated for the global error at final time $T$ with respect to number of meshpoints $n$. Symbol ✓ denotes convergence without a proven rate.

| Scheme | $f$ Riemann int. | $f$ BV | $f \in C^2$ |
|--------|:---:|:---:|:---:|
| Forward Euler | ✓ | $O(1/n)$ | $O(1/n)$ |
| Backward Euler | ✓ | $O(1/n)$ | $O(1/n)$ |
| Zero-order hold | ✓ | $O(1/n)$ | $O(1/n)$ |
| Approx. bilinear | ✓ | $O(1/n)$ | $O(1/n)$ |
| Bilinear | ✓ | $O(1/n)$ | $O(1/n^2)$ |

for $t \in [0, T]$, where $T > 0$ is some terminal time and $f : [0, T] \to \mathbb{R}$ is an input function. With specific choices of $A \in \mathbb{R}^{N \times N}$ and $B \in \mathbb{R}^{N \times 1}$, the solution $c : [0, T] \to \mathbb{R}^N$ encodes the continuous-time history of $f$ via $c_j(t) = \frac{1}{t} \left\langle f(\cdot), \sqrt{2j-1} P_{j-1}\left(\frac{2 \cdot}{t} - 1\right) \right\rangle_{L^2([0,t])} = \frac{\sqrt{2j-1}}{t} \int_0^t f(s) P_{j-1}\left(\frac{2s}{t} - 1\right) ds$, where $c_j(t)$ is the $j$-th component of $c(t)$, and $P_{j-1}$ is the $(j-1)$-th Legendre polynomial Gu et al. (2020). This compresses the otherwise intractable continuous-time history of $f$ into the single vector $c$, reducing the memory footprint to $\mathcal{O}(N)$. For instance, in language modeling, the model can condition on $c$ at each step rather than storing the entire dialogue history.

Unlike previous linear time-invariant (LTI) methods such as Legendre Memory Units (LMUs) Voelker et al. (2019), which consider measures with a fixed-length support, the LegS formulation considers the uniform measure on $[0, t]$ that widens with the progression of time $t$, and therefore provides a memory unit keeping track of the entire trajectory of $f(\cdot)$ from time 0 to $t$. This distinctive property makes the LegS formulation powerful in many practical applications.

However, despite receiving much attention for its use in state space models in deep learning, the careful mathematical foundation of the LegS ODE is missing. To begin with, the singularity at $t = 0$ renders the question of existence and uniqueness of the solution $c(t)$ a non-obvious matter. Moreover, the numerical methods used in the work of Gu et al. (2020) are not mathematically justified in the sense of convergence: In the limit of small stepsizes, do the discrete simulations converge to the true continuous-time solution? What regularity conditions must $f$ satisfy for such convergence?

**Contributions** In this work, we provide the rigorous mathematical foundations of the HiPPO-LegS ODE formulation and its discretization. Specifically, we show that (i) the solution to the LegS ODE exists and is unique, but the initial condition is fixed to a predetermined value depending on $f(0)$, (ii) the commonly used discretization schemes for LegS converge to the exact continuous-time solution for all Riemann integrable $f$, and (iii) obtain convergence rate guarantees. We summarize the convergence results in Table 1.

## 1.1 Related works

**State space models for deep learning** The use of state space models (SSMs) in deep learning has gained significant recent attention due to their ability to process sequential data efficiently. While the transformer architecture Vaswani et al. (2017) has become the standard for language models, recent SSM models such as Mamba Gu & Dao (2024) have been reported to achieve state-of-the-art results, especially in handling long sequences.

Large-scale SSMs deploy an initialization scheme motivated by the HiPPO theory Gu et al. (2023). One distinctive characteristic of state-of-the-art SSMs is that the computation cost displays a near-linear growth with respect to sequence length, unlike the quadratic growth of transformers. S4 Gu et al. (2022) uses the fast Fourier transform to attain the near-linear cost, whereas Mamba leverages hardware-aware computation techniques to attain near-linear parallel compute steps. The SSM architecture has been applied to or motivated numerous model structures Fu et al. (2023); Hasani et al. (2023); Sun et al. (2024); Peng et al. (2023) and is used across various modalities Zhu et al. (2024); Li et al. (2025); Shams et al. (2024).

**Legendre memory units for LSTMs** A fundamental challenge in training recurrent neural networks (RNNs) is the vanishing gradient problem, which causes long-range dependencies in temporal data to be lost during training Bengio et al. (1994); Le et al. (2015). While LSTMs Hochreiter & Schmidhuber (1997) alleviate this problem by incorporating nonlinear gating mechanisms, modeling very long sequences remains challenging. Motivated by applications in computational neuroscience, LMUs Voelker (2019); Voelker et al. (2019) introduced a novel approach to extend LSTM's capability to 'remember' the sequence information by constructing an $N$-dimensional ODE, for which the solution is the projection of the input function on the orthonormal basis of measure $\mathbf{1}_{[t-\theta,t]}$, where $\theta$ is a hyperparameter. The HiPPO framework could be understood as a generalization of LMUs. While LMU and its variants have proven to be effective for long sequence modeling Liu et al. (2024); Zhang et al. (2023); Chilkuri & Eliasmith (2021), their scope is limited to LTI methods.

**Convergence analysis of SSMs from control theory** State space models have been extensively studied in control theory Kalman (1960); Zabczyk (2020), with significant research dedicated to discretization schemes and their analysis Kowalczuk (1991). However, these results are not directly applicable to the LegS ODE due to their exclusive focus on LTI systems Karampetakis & Gregoriadou (2014) or their assumption of discrete-time inputs Meena & Janardhanan (2020). Furthermore, the objectives of state-space models in deep learning applications differ fundamentally from those in classical control theory, where for the latter, controlling or statistically estimating the state is usually the focus. This difference makes it challenging to directly adapt these results to modern deep-learning contexts.

## 2 Problem setting and preliminaries

In this work, we consider the LegS ODE

$$c'(t) = -\frac{1}{t}Ac(t) + \frac{1}{t}Bf(t)$$

for $t \in (0, T]$, where $T > 0$ is some terminal time and $c \colon [0, T] \to \mathbb{R}^N$ is the state vector encoding the continuous-time history of the input function $f \colon [0, T] \to \mathbb{R}$. The matrix $A \in \mathbb{R}^{N \times N}$ and vector $B \in \mathbb{R}^{N \times 1}$ are given by

$$A_{ij} = \begin{cases} (2i-1)^{1/2}(2j-1)^{1/2} & \text{if} \quad i > j \\ i & \text{if} \quad i = j \\ 0 & \text{if} \quad i < j, \end{cases} \qquad B_j = (2j-1)^{1/2}.$$

Since $A$ is lower-triangular with distinct diagonal entries, we immediately recognize that $A$ is diagonalizable with simple eigenvalues $\{1, 2, \ldots, N\}$. We denote the eigendecomposition as

$$A = VDV^{-1}, \qquad D = \mathrm{diag}\,(1, 2, \ldots, N).$$

with invertible $V \in \mathbb{R}^{N \times N}$. In the indexing of matrices and vectors, such as $A_{ij}$, $B_j$, and $c_i(t)$, we have $i, j \in \{1, \ldots, N\}$, i.e., we use 1-based indexing. (The prior HiPPO paper Gu et al. (2020) uses 0-based indexing.)

**Shifted Legendre polynomials** We write $P_j(x) \colon [-1, 1] \to [-1, 1]$ to denote the $j$-th Legendre polynomial, normalized such that $P_j(1) = 1$, for $j = 0, 1, \ldots$. However, we wish to operate on the domain $[0, 1]$, so we perform the change of variables $x \mapsto 2x - 1$. This yields

$$\tilde{P}_j(x) = P_j(2x-1) = \sum_{k=0}^{j}(-1)^j\binom{j}{k}\binom{j+k}{k}(-x)^k,$$

the $j$-th *shifted* Legendre polynomial, for $j = 0, 1, \ldots$. The shifted Legendre polynomials satisfy the recurrence relation

$$x\tilde{P}_j'(x) = j\tilde{P}_j(x) + \sum_{k=0}^{j-1}(2k+1)\tilde{P}_k(x), \tag{2}$$

which can be derived by combining the following well-known identities Arfken et al. (2011)

$$(2n+1)P_j(x) = P'_{j+1}(x) - P'_{j-1}(x), \qquad P'_{j+1}(x) = (n+1)P_j(x) + xP'_j(x).$$

**Numerical discretization methods**   In this work, we analyze the numerical methods of the LegS ODE used in the prior work Gu et al. (2020). For all the discretization methods, we consider a mesh grid with $n$ mesh points, with initial time $t_0 = 0$ and stepsize $h = T/n$. Starting from $c^0 = c(0)$, we denote $k$-th step of the numerical method as $c^k$, and $f(kh)$ as $f^k$.

The **backward Euler** method

$$c^{k+1} = \left(I + \frac{1}{k+1}A\right)^{-1}c^k + \left(I + \frac{1}{k+1}A\right)^{-1}\frac{1}{k+1}Bf^{k+1}$$

is well defined for $k = 0, 1, 2, \ldots, n-1$. However, the **forward Euler** method

$$c^{k+1} = \left(I - \frac{1}{k}A\right)c^k + \frac{1}{k}Bf^k$$

and the **bilinear (trapezoidal)** method

$$c^{k+1} = \left(I + \frac{1}{2(k+1)}A\right)^{-1}\left(I - \frac{1}{2k}A\right)c^k + \left(I + \frac{1}{2(k+1)}A\right)^{-1}\left(\frac{1}{2k}f^k + \frac{1}{2(k+1)}f^{k+1}\right)B$$

hold only for $k = 1, 2, \ldots, n-1$, and are not well defined for $k = 0$. One remedy would be to use the identity $c'(0) = (A+I)^{-1}Bf'(0)$, which we derive in Lemma 2. However, if $f$ is not differentiable at $t = 0$, then even this remedy is not possible. Hence, in Section 4, where we consider general $f$, we "zero-out" the ill-defined terms by setting them to be 0. So, for the **step 0 of forward Euler**, we set

$$c^1 = c^0$$

and for the **step 0 of bilinear (trapezoidal)**, we set

$$c^1 = \left(I + \frac{1}{2}A\right)^{-1}c^0 + \frac{1}{2}\left(I + \frac{1}{2}A\right)^{-1}Bf^1.$$

In the prior work Gu et al. (2020), the authors sidestep the division by 0 in the $1/k$ terms by shifting the $k$ index up by 1, leading to the **approximate bilinear** method

$$c^{k+1} = \left(I + \frac{1}{k+1}A/2\right)^{-1}\left(I - \frac{1}{k+1}A/2\right)c^k + \left(I + \frac{1}{k+1}A/2\right)^{-1}\left(\frac{1}{k+1}f^{k+1}\right)B$$

for $k = 0, 1, 2, \ldots, n-1$. In this work, we establish convergence of both the bilinear method (with zero-out) and the approximate bilinear method.

Lastly, prior work has also used the **Zero-order hold** method

$$c^{k+1} = e^{A\log\left(\frac{k}{k+1}\right)}c^k + A^{-1}\left(I - e^{A\log\left(\frac{k}{k+1}\right)}\right)Bf^k$$

for $k = 0, 1, 2, \ldots, n-1$, where we set $e^{A\log\left(\frac{n}{n+1}\right)} = 0$ at $n = 0$, consistent with the limit $e^{A\log\left(\frac{n}{n+1}\right)} \to 0$ as $n \to 0^+$. We also establish convergence for the zero-order hold method.

**Convergence of numerical discretization methods**   The numerical methods we consider are one-step methods of the form

$$c^{k+1} = c^k + h\Phi(t_k, t_{k+1}, c^k, c^{k+1}; h), \qquad k = 0, 1, \ldots, n-1$$

with stepsize $h = T/n$ and $t_k = t_0 + kh$ for $k = 0, \ldots, n-1$, approximating the solution to the initial value problem $c'(t) = g(t, c(t))$. Here, $\Phi$ is a numerical integrator making the approximation $\Phi(t_k, t_{k+1}, c^k, c^{k+1}; h) \approx c(t_{k+1}) - c(t_k) = \int_{t_k}^{t_{k+1}} g(s, c(s))\, ds$.

To analyze such methods, one often estimates the *local truncation error* (LTE) $T_k$ at timestep $t_k$ as $T_k = \frac{c(t_{k+1}) - c(t_k)}{h} - \Phi(t_k, t_{k+1}, c(t_k), c(t_{k+1}); h)$, and then estimates its accumulation to bound the *global error* $e_n = c(t_n) - c^n$, which is calculated at the endpoint. We say that a numerical discretization method is *convergent* if the global error converges to 0, i.e., if

$$\|c(t_n) - c^n\| \to 0, \qquad \text{as } n \to \infty.$$

Further, we quantify the *convergence rate* with the *order* of the method: we say the method has order $p$ if

$$\|c^n - c(t_n)\| \leq \mathcal{O}\left(1/n^p\right), \qquad \text{as } n \to \infty.$$

Classical ODE theory states that if the right-hand-side $g$ in the initial value problem is continuous with respect to $c$ and $t$, and Lipschitz continuous with respect to $c$, the solution exists and is unique in an interval including the initial point $t_0 = 0$. Moreover, under the same conditions, the global error can be bounded with the local truncation error Ascher & Petzold (1998); Süli & Mayers (2003). However, this standard theory does not apply to the LegS ODE due to the singularity at $t = 0$, and the non-smoothness of the input function $f$.

**Absolute continuity on a half-open interval** For $T \in (0, \infty)$, we say a function $c : (0, T] \to \mathbb{R}^N$ is absolutely continuous if its restriction to the closed interval $[\varepsilon, T]$ for any $\varepsilon \in (0, T)$ is absolutely continuous. (Recall that the standard definition of absolute continuity assumes a closed interval for the domain.) Even if $\lim_{t \to 0^+} c(t)$ is well defined and finite, the continuous extension of $c$ to $[0, T]$ may not be absolutely continuous on $[0, T]$. In other words, absolute continuity on $[\varepsilon, T]$ for all $\varepsilon \in (0, T)$ does not imply absolute continuity on $[0, T]$. We discuss this technicality further in Section 3, in the discussion following Theorem 1.

**Lebesgue point** Let $f : [0, T] \to \mathbb{R}$ be Lebesgue measurable and integrable. We say $f$ has a Lebesgue point at $t = 0$ if $\lim_{\varepsilon \to 0^+} \frac{1}{\varepsilon} \int_0^\varepsilon |f(s) - f(0)| \, ds = 0$. If $f(t)$ is continuous at $t = 0$, then $f$ has a Lebesgue point at $t = 0$.

## 3 LegS is well-posed

In this section, we show that the LegS ODE is well-posed despite the singularity. Crucially, however, we show that there is no freedom in choosing the initial condition.

**Theorem 1** (Existence and uniqueness). *For $T > 0$ and $c_0 \in \mathbb{R}^N$, we say $c : [0, T] \to \mathbb{R}^N$ is a solution (in the extended sense) of the LegS ODE if $c$ is continuous on $[0, T]$, absolutely continuous on $(0, T]$, $c$ satisfies equation 1 for almost all $t \in (0, T]$, and $c(0) = c_0$. Assume $f : [0, T] \to \mathbb{R}$ is Lebesgue measurable, integrable, and has a Lebesgue point at $t = 0$. Then, the solution exists and is unique if $c_0 = f(0)e_1$, where $e_1 \in \mathbb{R}^N$ is the first standard basis vector. Otherwise, if $c_0 \neq f(0)e_1$, a solution does not exist.*

*Proof.* Since $A$ is diagonalizable, the problem can be effectively decomposed into $N$ one-dimensional subproblems. Recall $A = VDV^{-1}$ where $D = \text{diag}\,(1, 2, \ldots, N)$. We see the LegS ODE equation 1 could be rewritten as

$$c'(t) = -\frac{1}{t}Ac(t) + \frac{1}{t}Bf(t) = -\frac{1}{t}VDV^{-1}c(t) + \frac{1}{t}Bf(t).$$

Multiply both sides by $V^{-1}$ and denote $\tilde{c}(t) = V^{-1}c(t)$. Then, the ODE becomes

$$\tilde{c}'(t) = V^{-1}c'(t) = -\frac{1}{t}DV^{-1}c(t) + \frac{1}{t}V^{-1}Bf(t) = -\frac{1}{t}D\tilde{c}(t) + \frac{1}{t}V^{-1}Bf(t),$$

which is a decoupled ODE with respect to $\tilde{c}$. Recalling $D_{jj} = j$, we see that the $j$-th component of the above equation is

$$\tilde{c}'_j(t) = -\frac{j}{t}\tilde{c}_j(t) + \frac{d_j}{t}f(t) \tag{3}$$

where $d_j = (V^{-1}B)_j$ and $\tilde{c}_j$ is the $j$-th component function of $\tilde{c}$. Since $V$ is a bijective linear map from $\mathbb{R}^N$ to $\mathbb{R}^N$, the existence and uniqueness of the solution of the LegS ODE is satisfied if and only if the existence and uniqueness of the solution of the ODE equation 3 is satisfied for all $j \in \{1, 2, ..., N\}$.

We now proceed by examining the existence and uniqueness of the solution of the ODE equation 3. We first establish existence by presenting the explicit form of the solution. Define $\tilde{c}_j : [0, T] \to \mathbb{R}$ as

$$\tilde{c}_j(t) = \begin{cases} \frac{d_j}{t^j} \int_0^t s^{j-1} f(s) ds & \text{if} \quad t \in (0, T] \\ \frac{d_j}{j} f(0) & \text{if} \quad t = 0. \end{cases} \tag{4}$$

By the fundamental theorem of calculus, $\frac{d}{dt} \left( \int_0^t s^{j-1} f(s) ds \right) = t^{j-1} f(t)$ holds for almost all $t \in (0, T]$ and thus $\tilde{c}_j$ is differentiable for almost all $t \in (0, T]$. Therefore,

$$t^j \left( \tilde{c}_j'(t) + \frac{j}{t} \tilde{c}_j(t) \right) = \frac{d}{dt} (t^j \tilde{c}_j(t)) = \frac{d}{dt} \left( d_j \int_0^t s^{j-1} f(s) ds \right) = d_j t^{j-1} f(t)$$

holds for almost all $t \in (0, T]$. Dividing both sides by $t^j$, we conclude that $\tilde{c}_j$ satisfies equation 3 for almost all $t \in (0, T]$.

We now show $\tilde{c}_j$ is continuous on $[0, T]$. It is sufficient to check $\tilde{c}_j$ is continuous at $t = 0$ by showing $\lim_{t \to 0} \frac{d_j}{t^j} \int_0^t s^{j-1} f(s) ds = \frac{d_j}{j} f(0)$. Since $f$ is locally integrable and has a Lebesgue point at $t = 0$, observe that

$$0 = \lim_{t \to 0^+} \frac{1}{t} \int_0^t |f(s) - f(0)| ds = \lim_{t \to 0^+} \int_0^1 |f(ts) - f(0)| ds,$$

where the second equality follows by change of variables $x = s/t$. Therefore, we could deduce

$$\left| \lim_{t \to 0^+} \frac{d_j}{t^j} \int_0^t s^{j-1} f(s) ds - \frac{d_j}{j} f(0) \right| = \lim_{t \to 0^+} \frac{|d_j|}{t^j} \left| \int_0^t s^{j-1} (f(s) - f(0)) ds \right|$$

$$\leq \lim_{t \to 0^+} |d_j| \int_0^t |f(ts) - f(0)| \, ds \longrightarrow 0.$$

concluding that $\tilde{c}_j$ is continuous at $t = 0$. Lastly, $\tilde{c}_j$ is absolutely continuous on $(0, T]$, since for every $[t_0, t] \subset (0, T]$, both $\frac{1}{t^j}$ and $\int_0^t s^{j-1} f(s) ds$ are absolutely continuous on $[t_0, t]$ and therefore their product is also absolutely continuous on $[t_0, t]$. Hence we conclude that $\tilde{c}_j$ is a solution of the ODE equation 3.

We now establish uniqueness. Suppose $\hat{c}_j$ is another solution of the ODE equation 3. Multiplying both sides of equation 3 by $t^j$ and reorganizing, for almost all $t \in (0, T]$ we have

$$\frac{d}{dt} (t^j \hat{c}_j(t)) = t^j \left( \hat{c}_j'(t) + \frac{j}{t} \hat{c}_j(t) \right) = d_j t^{j-1} f(t).$$

Since $\hat{c}_j$ is a solution, it is absolutely continuous on $(0, T]$, therefore $t^j \hat{c}_j(t)$ is absolutely continuous on $(0, T]$. Thus for $[t_0, t] \subset (0, T]$, by the fundamental theorem of calculus we obtain

$$t^j \hat{c}_j(t) - t_0^j \hat{c}_j(t_0) = d_j \int_{t_0}^t s^{j-1} f(s) ds.$$

Since $\hat{c}_j$ is a solution, it is continuous at 0, and we obtain $t^j \hat{c}_j(t) = d_j \int_0^t s^{j-1} f(s) ds$ by taking limit $t_0 \to 0^+$. Dividing both sides by $t^j$ we conclude

$$\hat{c}_j(t) = \frac{d_j}{t^j} \int_0^t s^{j-1} f(s) ds = \tilde{c}_j(t)$$

for all $t \in (0, T]$. It remains to check $\hat{c}_j(0) = \tilde{c}_j(0)$. Since $\hat{c}$ is continuous at 0, we know $\hat{c}_j(0) = \lim_{t \to 0^+} \hat{c}_j(t)$. Thus

$$\hat{c}_j(0) = \lim_{t \to 0^+} \hat{c}_j(t) = \lim_{t \to 0^+} \tilde{c}_j(t) = \tilde{c}_j(0).$$

Therefore, we conclude $\hat{c}_j(t) = \tilde{c}_j(t)$ for all $t \in [0, T]$, the solution of the ODE equation 3 is unique.

As a result, we conclude the solution of the LegS ODE uniquely exists if $\tilde{c}_j(0) = \frac{d_j}{j} f(0)$, and it is given by $c = V\tilde{c}$. Finally, we show the unique solution $c = V\tilde{c}$ should satisfy $c(0) = f(0)e_1$. From

$$V^{-1}c_j(0) = \tilde{c}_j(0) = \frac{d_j}{j} f(0) = \frac{1}{j}(V^{-1}B)_j f(0),$$

we see

$$V^{-1}c(0) = \mathrm{diag}\left(1, \frac{1}{2}, \frac{1}{3}, ..., \frac{1}{N}\right)(V^{-1}B)f(0) = D^{-1}V^{-1}Bf(0).$$

Multiplying both sides by $V$, we conclude

$$c(0) = VD^{-1}V^{-1}Bf(0) = (VDV^{-1})^{-1}Bf(0) = A^{-1}Bf(0) = f(0)e_1$$

where $e_1 = [1, 0, \ldots, 0]^t$. Therefore if $c_0 = f(0)e_1$, the solution exists and is unique, and otherwise, there is no solution. $\qquad\square$

**Remark 3.1.** *Theorem 1 does not guarantee that $c$ is absolutely continuous on the closed interval $[0, T]$, only on the half-open interval $(0, T]$. The following lemma provides a counterexample of a continuous input function $f$ such that the corresponding solution of the LegS ODE is not absolutely continuous on $[0, T]$.*

**Lemma 1.** *Let $T = 1/2$. Consider the LegS ODE with $f: [0, T] \to \mathbb{R}$ defined as*

$$f(t) = \begin{cases} \frac{d}{dt}\left(\frac{t^2}{\log(1/t)} \sin\left(1/t\right)\right) & \text{if } 0 < t \leq 1/2 \\ 0 & \text{otherwise.} \end{cases}$$

*Since $f$ is continuous on $[0, T]$ (including at $t = 0$) it satisfies the conditions of Theorem 1. However, the solution of the LegS ODE is not absolutely continuous on $[0, T]$.*

*Proof Sketch.* We prove this by showing that the first component of $c(t)$ corresponding to the given input function $f(t)$ is not absolutely continuous. The full proof can be found in appendix A. $\qquad\square$

**Remark 3.2.** *Recall that the motivation of the LegS ODE is to provide an online approximation of the input function $f$. By change of variables, $\{\frac{\sqrt{2j-1}}{t} P_{j-1}\left(\frac{2s}{t} - 1\right)\}_{j \in \mathbb{N}}$ could be shown to be an orthogonal basis on the interval $[0, t]$, with respect to the $L^2([0, t])$ norm. The following corollary shows that the solution found in Theorem 1 is the projection of $f$ onto this basis. We defer the proof to appendix B.*

**Corollary 1.** *The solution $c$ of the LegS ODE as defined in Theorem 1 is, if it exists, an $L^2$-approximation of $f$ on $\frac{1}{t}\mathbf{1}_{[0,t]}$ for all $t \in (0, T]$ in the sense that the $j$-th component of $c(t) \in \mathbb{R}^N$ is given by*

$$c_j(t) = \frac{1}{t}\left\langle f(\cdot), \sqrt{2j-1} P_{j-1}\left(\frac{2\cdot}{t} - 1\right)\right\rangle_{L^2([0,t])} = \frac{\sqrt{2j-1}}{t}\int_0^t f(s)P_{j-1}\left(\frac{2s}{t} - 1\right) ds, \tag{5}$$

*for all $t \in (0, T]$, where $P_{j-1}$ denotes the $(j-1)$-th Legendre polynomial.*

**Remark 3.3.** *Regarding stability, using the exact projection characterizations from Corollary 1, we could obtain boundedness and continuous dependence on the input for finite horizons. Boundedness follows directly from*

$$|c_j(t)| \leq \sqrt{2j-1} \sup_{s \in [0,t]} |f(s)|$$

*on any fixed $t \in [0, T]$. Now for continuous dependence on data, let $c^f$ and $c^g$ be the solutions corresponding to inputs $f$ and $g$. Again by Corollary 1, for $t > 0$,*

$$|c_j^f(t) - c_j^g(t)| \leq \frac{\sqrt{2j-1}}{t}\int_0^t |f(s) - g(s)|\left|P_{j-1}\left(\frac{2s}{t} - 1\right)\right| ds$$

$$\leq \frac{\sqrt{2j-1}}{t}\int_0^t |f(s) - g(s)| ds \leq \sqrt{2j-1}\|f - g\|_{L^\infty([0,T])}.$$

*At $t = 0$, $|c^f(0) - c^g(0)| = |f(0) - g(0)||e_1| \leq \|f - g\|_{L^\infty([0,T])}$. Hence we have Lipschitz continuity*

$$\|c_j^f - c_j^g\|_{L^\infty([0,T])} \leq \sqrt{2j-1}\|f - g\|_{L^\infty([0,T])}$$

*for all $j \in \{1, 2, \ldots, N\}$.*

**Remark 3.4.** *The well-posedness argument of Theorem 1 crucially relies on the fact that all eigenvalues of $A$ are positive. To see what happens when $A$ has negative eigenvalues, consider the case $N = 1$ and $A = -1$. This leads to the ODE*

$$\frac{d}{dt}c(t) = \frac{1}{t}c(t) + \frac{1}{t}f(t), \qquad c(0) = c_0,$$

*for which $c(t) = t\int_0^t \frac{1}{s^2}f(s)\,ds + Ct$ is a solution for any $C \in \mathbb{R}$. Since the initial condition does not determine the value of $C$, the solution is not unique.*

**Remark 3.5.** *If a stronger condition, such as the (one-sided) differentiability of $f(t)$ at $t = 0$ is provided, the derivative of $c(t)$ at $t = 0$ can be calculated as in the following lemma. In setups where $f'$ is available, this identity could be used to implement the first iteration of the forward Euler method and the bilinear method.*

**Lemma 2** (Behavior at $t = 0$). *Consider the setup of Theorem 1, and further assume that $f'(0) := \lim_{t\to 0+} \frac{f(t)-f(0)}{t}$ exists. Then, $c'(0) := \lim_{t\to 0+} \frac{c(t)-c(0)}{t}$ exists and*

$$c'(0) = (A + I)^{-1}Bf'(0).$$

*Proof.* We examine the differentiability of $V^{-1}c(t) = \tilde{c}(t) = (\tilde{c}_j(t))_{j=1}^N$ by checking for each component. Recall that for the $j$-th component we have $\tilde{c}_j(t) = \frac{d_j}{t^j}\int_0^t s^{j-1}f(s)ds$ and $\tilde{c}_j(0) = \frac{d_j}{j}f(0)$ from equation 4. Then,

$$\lim_{t\to 0+}\frac{\tilde{c}_j(t) - \tilde{c}_j(0)}{t} = \lim_{t\to 0+}\frac{d_j}{t}\left(\int_0^1 x^{j-1}f(tx)dx - \frac{1}{j}f(0)\right)$$

$$= \lim_{t\to 0+}\int_0^1 x^j\frac{f(tx)-f(0)}{tx}dx \overset{(1)}{=} \frac{d_j}{j+1}f'(0).$$

where DCT was used for (1). Folding back to vector form, recalling $(V^{-1}c'(0))_j = \tilde{c}_j'(t) = \frac{d_j}{j+1}f'(0) = \frac{1}{j+1}(V^{-1}B)_jf'(0)$, we obtain

$$V^{-1}c'(0) = \text{diag}\left(\frac{1}{2}, \frac{1}{3}, ..., \frac{1}{N+1}\right)(V^{-1}B)f'(0)$$

and hence $c'(0) = (A + I)^{-1}Bf'(0)$. Note that $(A+I)^{-1}B = [1/2, 1/(2\sqrt{3}), 0, ..., 0]^t$. $\qquad\square$

## 4 Convergence of LegS discretization schemes

In this section, we address the convergence of the numerical discretization methods introduced in Section 2, i.e., do the methods produce numerical solutions $c^n$ that converge to the exact continuous-time solution $c(t)$ as $h \to 0$?

As we discuss in Section 4.1, the standard analysis based on local truncation error does not lead to a convergence guarantee for all of the schemes under consideration, and such approaches would require certain local regularity conditions on $f$, such as (Lipschitz) continuity. Rather, in Section 4.2, we identify the numerical schemes as quadrature rules on the input function $f$. Using this insight, in Section 4.3, we show that the discretization schemes are convergent under the general assumption of Riemann integrability of $f$.

Extending the framework to accommodate general Riemann integrable functions $f$ is important, given the nature of the application. The HiPPO memory unit is used in deep learning to analyze sequence data, such as language or audio signals Gu et al. (2020). For such data, there is no inherent expectation of smoothness, and discontinuities are to be expected. Therefore, we aim to guarantee that the mathematics remains sound for such data.

### 4.1 Convergence for smooth $f$

Discretization methods of ODEs with well-behaved right-hand-sides have a well-established theory based on the local truncation error (LTE), and for sufficiently smooth input function $f$, the standard techniques can be applied to the LegS ODE despite the singularity at $t = 0$. For example, it can be shown that LTE for the forward Euler method applied to the LegS ODE satisfies

$$|T_k| \leq \frac{1}{2}hM_2, \qquad M_2 = \max_{t\in[0,T]} |c''(t)|.$$

However, when $f$ is not differentiable, then $c''(t)$ may not be bounded, and this approach, as is, fails to yield a convergence guarantee. Another issue is that the LTE for approximate bilinear method does not converge to 0, even for smooth $f$. For $N = 1$, the approximate bilinear method reduces to

$$c^{k+1} = \frac{2k+1}{2k+3}c^k + \frac{2}{2k+3}f^{k+1} = c^k - \frac{2}{2k+3}(c^k - f^{k+1}).$$

Using the exact solution $c(t) = \frac{1}{t}\int_0^t f(s)ds$, the exact value of the LTE is

$$hT_k = c(t_{k+1}) - c(t_k) + \frac{2}{2k+3}(c(t_k) - f^{k+1})$$

$$= \frac{1}{(k+1)h}\int_0^{(k+1)h} f(s)ds - \frac{2k+1}{k(2k+3)h}\int_0^{kh} f(s)ds - \frac{2}{2k+3}f^{k+1}.$$

With the linear function $f(x) = ax$ $(a \neq 0)$ as a particular choice, we obtain that at step $k = 0$,

$$T_0 = \frac{a}{2} - \frac{2a}{3} = -\frac{a}{6} \neq 0.$$

Thus, the LTE of the approximate bilinear method does not vanish as $h \to 0$. Consequently, a naive global error analysis based on the LTE will not guarantee convergence. In Section 4.3, we employ an alternative proof technique to establish convergence.

### 4.2 LegS discretizations are quadratures of $f$

In this section, we provide the key insight that we can identify the discretization methods applied to the solution $c$ as quadrature rules on the input function $f$. Recall that the LegS ODE was proposed for online approximation of the input function $f(t)$ on the interval $[0, t]$. Specifically, Corollary 1 says that the $j$-th component of the solution is given by

$$c_j(t) = \frac{\sqrt{2j-1}}{t}\int_0^t P_{j-1}\left(\frac{2s}{t} - 1\right)f(s)ds,$$

where $P_{j-1}$ denotes the $(j-1)$-th Legendre polynomial, for $j = 1, 2, \ldots, N$. So $c_j(t)$ is a (signed) weighted integral of $f(\cdot)$ on $[0, t]$. The following lemma shows that the numerical schemes of Section 2 can, in fact, be interpreted as quadrature rules with uniformly spaced nodes, and in Section 4.3, we show that these quadratures approximate the integral.

**Lemma 3.** *Consider applying any of the discretization methods introduced in Section 2 (forward Euler, backward Euler, bilinear, approximate bilinear, or zero-order hold) to the LegS ODE equation 1, with initial time $t_0 = 0$ and timestep $h = T/n$ with $n \geq 2$. Then, the numerical solution $c^n$ at step $n$ can be expressed in the form*

$$c^n = \frac{1}{n}\sum_{l=0}^{n} \alpha_l^{(n)} f^l$$

*for some $\alpha_l^{(n)} \in \mathbb{R}^N$ that depend only on $l$ and $n$, where $f^l = f(lh)$.*

*Proof Sketch.* The proof consists of direct computation and induction. For notational simplicity, we define $Q_n$ for $n \geq 2$, and $\tilde{Q}_n$, $R_n$, $\tilde{R}_n$ for $n \geq 1$ as follows:

$$Q_n := \prod_{j=1}^{n-1} \left( I - \frac{1}{j+1} A \right), \qquad \tilde{Q}_n := \prod_{j=1}^{n} \left( I - \frac{1}{2j} A \right),$$

$$R_n := \prod_{j=1}^{n} \left( I + \frac{1}{j} A \right)^{-1}, \qquad \tilde{R}_n := \prod_{j=1}^{n} \left( I + \frac{1}{2j} A \right)^{-1}.$$

Then, the numerical solution $c^n$ obtained by applying forward Euler, backward Euler, bilinear, approximate bilinear, and zero-order hold are computed as follows:

$$(c^n)_{\text{for}} = Q_{n-1}(e_1 f^0 + B f^1) + Q_{n-1} \sum_{l=2}^{n-1} \frac{1}{l} Q_l^{-1} B f^l \tag{6}$$

$$(c^n)_{\text{back}} = R_n(e_1 f^0 + B f^1) + R_n \sum_{l=1}^{n-1} \frac{1}{l+1} R_l^{-1} B f^{l+1} \tag{7}$$

$$(c^n)_{\text{bilin}} = \tilde{Q}_{n-1}\tilde{R}_n e_1 f^0 + \tilde{Q}_{n-1}\tilde{R}_n \sum_{l=1}^{n-1} \frac{1}{2l} (\tilde{Q}_l^{-1}\tilde{R}_l^{-1} + \tilde{Q}_{l-1}^{-1}\tilde{R}_{l-1}^{-1}) B f^l + \tilde{R}_n \tilde{R}_{n-1}^{-1} \frac{B}{2n} f^n \tag{8}$$

$$(c^n)_{\text{approx}} = \tilde{Q}_n \tilde{R}_n c^0 + \tilde{Q}_n \tilde{R}_n \sum_{l=1}^{n-1} \tilde{Q}_{l+1}^{-1}\tilde{R}_l^{-1} \left( \frac{1}{l+1} B f^{l+1} \right) \tag{9}$$

$$(c^n)_{\text{zoh}} = \frac{1}{n} \sum_{l=0}^{n} \alpha_l^{(n)} f(lh) = \sum_{l=0}^{n-1} \left( e^{A \log\left(\frac{l+1}{n}\right)} - e^{A \log\left(\frac{l}{n}\right)} \right) A^{-1} B f^l. \tag{10}$$

The full derivation can be found in appendix C. With some inspection, one can conclude that all of the numerical solutions above acquire the desired form. $\qquad\square$

### 4.3 Convergence for Riemann integrable $f$

In this section, we prove the convergence of all discretization methods of interest for Riemann integrable $f$'s. In particular, we prove the convergence of the approximate bilinear method, justifying its use for the experiments in the HiPPO paper Gu et al. (2020). The results are summarized in the following theorem:

**Theorem 2** (Convergence of discretization schemes for Riemann integrable $f$). *Consider the LegS equation equation 1 with dimension $N \geq 1$ and domain $t \in [0, T]$, where $T > 0$. Assume $f$ is Riemann integrable on $[0, T]$. Let $n \geq 2$ be the number of mesh points and let $h = T/n$. Consider discretization methods with initialization and iterations*

$$c^0 = f(0)e_1, \qquad c^{k+1} = c^k + h\Phi(t_k, t_{k+1}, c^k, c^{k+1}; h)$$

*using mesh points $t_k = kh$ for $k = 0, 1, 2, \ldots, n$, where $\Phi$ is the one-step integrator defined by the given discretization method. Denote the exact solution at step $n$ as $c(t_n) = c(nh) \in \mathbb{R}^N$. Then, for all the forward Euler, backward Euler, bilinear, approximate bilinear, and zero-order hold methods defined in Section 2, we have convergence of the numerical solution to the exact solution in the sense that*

$$\|c^n - c(t_n)\| \to 0, \qquad as \quad n \to \infty.$$

To prove the theorem above, note that in light of Lemma 3, we can denote the iterates of the numerical schemes as $c^n = \frac{1}{n} \sum_{l=0}^{n} \alpha_l^{(n)} f^l$. If we can show

$$c_j^n = \frac{1}{n} \sum_{l=0}^{n} \left( \alpha_l^{(n)} \right)_j f^l \quad \longrightarrow \quad c_j(t_n) = \frac{\sqrt{2j-1}}{t_n} \int_0^{t_n} P_{j-1} \left( \frac{2s}{t_n} - 1 \right) f(s) ds,$$

as $n \to \infty$, for all $j \in \{1, 2, \ldots, N\}$ and $P_j$ is the $j$-th Legendre polynomial, we are done. Now, the key idea of the proof is, instead of directly characterizing the coefficients $\alpha_l^{(n)}$, to consider a function sequence defined on $[0, 1]$ that interpolates those points. This significantly reduces the complexity of analyzing the asymptotic behavior of the numerical solution as the number of mesh points $n$ goes to infinity. We start with an elementary lemma that enables this approach.

**Lemma 4.** *Let $f\colon [0, t] \to \mathbb{R}$ be a Riemann integrable function. Let $\{G^{(n)}\}_{n \in \mathbb{N}}$ be a sequence of continuous functions defined on $[0, 1]$ uniformly converging to $G \in C[0, 1]$. Then, for $h = t/n$,*

$$\frac{1}{n} \sum_{l=1}^{n} G^{(n)} \left(\frac{l}{n}\right) f(lh) \to \frac{1}{t} \int_0^t G\left(\frac{s}{t}\right) f(s) ds \qquad as \quad n \to \infty.$$

*Proof.* Fix $\epsilon > 0$. Since $f$ is Riemann integrable and $G$ is continuous, $G\left(\frac{s}{t}\right) f(s)$ is Riemann integrable for $s \in [0, t]$. Hence we can find $N_1 \in \mathbb{N}$ such that for all $n \geq N_1$, $\left| \frac{1}{n} \sum_{l=1}^{n} G\left(\frac{l}{n}\right) f(lh) - \frac{1}{t} \int_0^t G\left(\frac{s}{t}\right) f(s) ds \right| < \epsilon/2$. Since $f$ is bounded, $\sup_{x \in [0,t]} |f(x)| \leq M$ holds for some $M > 0$. Then, due to the uniform convergence of $G^{(n)}$, we can find $N_2 \in \mathbb{N}$ such that $\|G - G^{(n)}\| < \frac{\epsilon}{2M}$ holds for all $n \geq N_2$. Therefore, choosing $n \in \mathbb{N}$ with $n \geq \max\{N_1, N_2\}$, we conclude:

$$\left| \frac{1}{t} \int_0^t G\left(\frac{s}{t}\right) f(s) ds - \frac{1}{n} \sum_{l=1}^{n} G^{(n)} \left(\frac{l}{n}\right) f(lh) \right|$$
$$\leq \left| \frac{1}{t} \int_0^t G\left(\frac{s}{t}\right) f(s) ds - \frac{1}{n} \sum_{l=1}^{n} G\left(\frac{l}{n}\right) f(lh) \right| + \frac{1}{n} \sum_{l=1}^{n} \left| G\left(\frac{l}{n}\right) f(lh) - G^{(n)} \left(\frac{l}{n}\right) f(lh) \right|$$
$$\leq \frac{\epsilon}{2} + M\|G - G^{(n)}\|_{\sup} \leq \epsilon.$$

$\square$

This result is relevant since if we interpret the shifted Legendre polynomials as $G\left(\frac{s}{t}\right)$ in the lemma above, we can obtain a sufficient condition for a numerical solution to converge to the exact solution of the ODE. This observation is specified in the next corollary.

**Corollary 2.** *Consider an array of vectors $\left\{ (c^n)_j = \frac{1}{n} \sum_{l=0}^{n} \left(\alpha_l^{(n)}\right)_j f(lh) \in \mathbb{R}^N \right\}_{n \in \mathbb{N}}$, where $h = t/n$ and $f\colon [0, t] \to \mathbb{R}$ is a Riemann integrable function. Assume that for each $j$, there exists a vector-valued degree $j - 1$ polynomial function sequence $\{F^{(n)}\colon [0, 1] \to \mathbb{R}^N\}_{n \in \mathbb{N}}$ satisfying the following condition*

$$\left(F\left(\frac{l}{n}\right)\right)_j = \left(\alpha_l^{(n)}\right)_j, \qquad \forall l \in \{1, 2, \ldots, n-1\}. \tag{11}$$

*and $\left\| \left(F^{(n)}(\cdot)\right)_j - \sqrt{2j-1} P_{j-1}(2 \cdot -1) \right\|_{\sup([0,1])} \to 0$ as $n \to \infty$ for all $j \in \{1, 2, \ldots, N\}$. Moreover assume that the endpoint coefficients are uniformly bounded:*

$$\sup_n |(\alpha_0^{(n)})_j| < \infty, \qquad \sup_n |(\alpha_n^{(n)})_j| < \infty.$$

*Then,*

$$\|c^n - c(t_n)\| \to 0, \qquad as \quad n \to \infty.$$

*Proof.* Since the sequence $\{F^{(n)}\}$ is a polynomial sequence defined on a compact domain, with fixed degree of order, it is uniformly bounded. Hence, for all $j \in \{1, 2, \ldots, N\}$, we can choose some $B_j > 0$ and $C_j > 0$ such that $F_j \leq B_j$ for all $F \in \{F^{(k)}\}$ and $\sup_{n,l} \left(\alpha_l^{(n)}\right)_j \leq C_j$, where for the last inequality, we used the uniform boundedness assumption for the endpoint coefficients. Also, $\sup_{x \in [0,t]} |f(x)| \leq M$ for some $M > 0$ from the definition of Riemann integrable functions. Fix component index $j$.

Since $\left(F^{(k)}\left(\frac{l}{n}\right)\right)_j = \left(\alpha_l^{(n)}\right)_j$ for $l \in \{1, 2, \ldots, n-1\}$, we can write

$$(c^n)_j = \frac{1}{n} \sum_{l=1}^{n} \left(F^{(n)}\left(\frac{l}{n}\right)\right)_j f(lh) + \frac{1}{n}\left(\left(\alpha_0^{(n)}\right)_j f(0) + \left(\alpha_n^{(n)}\right)_j f(nh) - \left(F^{(n)}(1)\right)_j f(nh)\right).$$

Fix $\epsilon > 0$. Since the function sequence $F^{(n)}$ satisfies the conditions in Lemma 4, we can find $N_1 \in \mathbb{N}$ such that $\left|\frac{1}{n}\sum_{l=1}^{n}\left(F^{(n)}\left(\frac{l}{n}\right)\right)_j f(lh) - \frac{\sqrt{2j-1}}{t}\int_0^t P_{j-1}\left(\frac{2s}{t}-1\right)f(s)ds\right| < \epsilon/2$ holds for all $n \geq N_1$. Choose $n \in \mathbb{N}$ so that $n \geq \max\{N_1, \frac{2M(2C_j+B_j)}{\epsilon}\}$. Constructing the following triangle inequality, we conclude the following holds for all $j \in \{1, 2, \ldots, N\}$:

$$\left|(c^n)_j - \frac{\sqrt{2j-1}}{t}\int_0^t P_{j-1}\left(\frac{2s}{t}-1\right)f(s)ds\right|$$

$$\leq \left|\frac{1}{n}\sum_{l=1}^{n}\left(F^{(n)}\left(\frac{l}{n}\right)\right)_j f(lh) - \frac{\sqrt{2j-1}}{t}\int_0^t P_{j-1}\left(\frac{2s}{t}-1\right)f(s)ds\right|$$

$$+ \frac{1}{n}\left|\left(\alpha_0^{(n)}\right)_j f(0) + \left(\alpha_n^{(n)}\right)_j f(nh)\right| + \frac{1}{n}\left|\left(F^{(n)}(1)\right)_j f(nh)\right| \leq \frac{\epsilon}{2} + \frac{M}{n}(2C_j + B_j) \leq \epsilon$$

$\square$

The result of this corollary implies that instead of directly proving the convergence of the numerical solution $c^n$ to the exact solution, it would suffice to find a function sequence $\{F^{(n)}\}_{n\in\mathbb{N}}$ satisfying equation 11 that converges to the (scaled) shifted Legendre polynomial. A natural choice to construct such a sequence would be to interpolate the $n-1$ points using polynomials so that the function would satisfy equation 11. However, the degree of the interpolating polynomials in the sequence could diverge as $n \to \infty$, complicating the analysis of their limiting behavior. Surprisingly, the following lemma shows that for all discretization methods of interest, the sequence $\{F^{(n)}\}_{n\in\mathbb{N}}$ is a polynomial sequence of fixed degree. Note that if we interpolate $n+1$ points, i.e. including $\left(0, \left(\alpha_0^{(n)}\right)_j\right), \left(1, \left(\alpha_n^{(n)}\right)_j\right)$ as interpolating points, the following lemma does not work.

**Lemma 5.** *Denote the $j$-th index of the numerical solution obtained by a given discretization method as $(c^n)_j = \frac{1}{n}\sum_{l=0}^{n}\left(\alpha_l^{(n)}\right)_j f^l$ where $h = t/n$. Consider the following $n-1$ points*

$$\left(\frac{1}{n}, \left(\alpha_1^{(n)}\right)_j\right), \left(\frac{2}{n}, \left(\alpha_2^{(n)}\right)_j\right), \ldots, \left(1-\frac{1}{n}, \left(\alpha_{n-1}^{(n)}\right)_j\right). \tag{12}$$

*Then, for all $n \geq 2$ and $j \in \{1, 2, \ldots, N\}$, there exists a polynomial $F_j^{(n)}$ of degree at most $j-1$ that interpolates the above $n-1$ points obtained by any discretization method introduced in Section 2. Moreover, define $\{F^{(n)}(x)\}_{n\in\mathbb{N}}$ as a vector-valued function sequence such that for each $n$, the $j$-th component is $F_j^{(n)}(x)$. Then, given the eigenvalue decomposition $A = VDV^{-1}$, the sequence $\{F^{(n)}(x)\}_{n\in\mathbb{N}}$ pointwise converges to*

$$F(x) = V\,\mathrm{diag}\left(1, 2x, 3x^2, \ldots, Nx^{N-1}\right)V^{-1}e_1$$

*as $n \to \infty$ for every fixed $x \in [0, 1]$.*

*Proof.* Here we provide the proof for the forward Euler case, and defer the proof for the other methods to appendix D. Fix $n \in \mathbb{N}$. Let $p^{(n)}: [0,1] \to \mathbb{R}^N$ be some vector whose $j$-th component is a function interpolating the $n-1$ points in equation 12. Then, referring to equation 6, $p^{(n)}$ by construction satisfies

$$p^{(n)}\left(\frac{l}{n}\right) = \frac{n}{l}\prod_{k=l+1}^{n-1}\left(I - \frac{1}{k}A\right)Ae_1$$

for $l \in \{1, 2, ..., n-1\}$. Let $A = VDV^{-1}$ where $D \in \mathbb{R}^{N \times N}$ is the diagonal matrix with entries $(D)_{jj} = j$. Then,

$$\prod_{k=l+1}^{n-1} \left( I - \frac{1}{k} A \right) = \prod_{k=l+1}^{n-1} V \left( I - \frac{1}{k} D \right) V^{-1} = V \left( \prod_{k=l+1}^{n-1} \left( I - \frac{1}{k} D \right) \right) V^{-1}$$

$$= V \mathrm{diag} \left( \prod_{k=l+1}^{n-1} \left( 1 - \frac{1}{k} \right), \ldots, \prod_{k=l+1}^{n-1} \left( 1 - \frac{j}{k} \right), \ldots, \prod_{k=l+1}^{n-1} \left( 1 - \frac{N}{k} \right) \right) V^{-1}.$$

Now, since we are interested in the limiting behavior of $n \to \infty$, we can assume that $n$ is considerably larger than $N$. Then we can cancel out terms in the denominator and the numerator to calculate the $i$-th term in the diagonal matrix,

$$\frac{1}{l} \prod_{k=l+1}^{n-1} \left( 1 - \frac{i}{k} \right) = \frac{1}{l} \frac{1}{\prod_{k=l+1}^{n-1} k} \prod_{k=l+1}^{n-1} (k - i) = \frac{\prod_{k=1}^{i-1} (l - k)}{\prod_{k=1}^{i} (n - k)}$$

where $\prod_{k=1}^{0} (l - k) = 1$. Therefore we arrive at

$$p^{(n)} \left( \frac{l}{n} \right) = nV \mathrm{diag} \left( \frac{1}{n-1}, \frac{(l-1)}{(n-1)(n-2)}, \ldots, \frac{\prod_{k=1}^{N-2} (l-k)}{\prod_{k=1}^{N-1} (n-k)}, \frac{\prod_{k=1}^{N-1} (l-k)}{\prod_{k=1}^{N} (n-k)} \right) DV^{-1} e_1$$

for $l \in \{1, 2, \ldots, n-1\}$. Now we change variables and let $l = nx$. Then, we can define a vector function $F^{(n)}$ with the $j$-th component

$$F_j^{(n)}(x) = ne_j^t V \mathrm{diag} \left( \frac{1}{n-1}, \frac{(nx-1)}{(n-1)(n-2)}, \ldots, \frac{\prod_{k=1}^{N-2} (nx-k)}{\prod_{k=1}^{N-1} (n-k)}, \frac{\prod_{k=1}^{N-1} (nx-k)}{\prod_{k=1}^{N} (n-k)} \right) DV^{-1} e_1.$$

such that $F^{(n)} \left( \frac{l}{n} \right) = p^{(n)} \left( \frac{l}{n} \right)$ for all $l \in \{1, 2, \ldots, n-1\}$, for all $x \in [0, 1]$. Notice in the above expression of $F_j^{(n)}(x)$ that the $i$-th term in the diagonal matrix is an $(i-1)$-degree polynomial of $x$. Since $V$ and $V^{-1}$ are both lower triangular, we conclude that $F_j^{(n)}$ is a $(j-1)$-degree polynomial interpolating the $n-1$ points of interest. Moreover, for any fixed $x \in [0, 1]$, taking the limit $n \to \infty$ yields

$$\lim_{n \to \infty} F^{(n)}(x) = V \mathrm{diag} \left( 1, 2x, 3x^2, \ldots, Nx^{N-1} \right) V^{-1} e_1. \tag{13}$$

$\square$

Combining the results, we can prove Theorem 2.

*Proof of Theorem 2.* Given a discretization method, we can express the numerical solution as $c^n = \frac{1}{n} \sum_{l=0}^{n} \alpha_l^{(n)} f^l$ by Lemma 3. Then, define the function sequence $\{F^{(n)}\}_{n \in \mathbb{N}}$ as in Lemma 5. Then for all $n \in \mathbb{N}$, the $j$-th component of $F^{(n)} : [0, 1] \to \mathbb{R}^N$ is an at-most-$(j-1)$-degree polynomial, i.e., $\left( F^{(n)} \right)_j \in \mathcal{P}_{j-1}$, where $\mathcal{P}_j$ is the space of polynomials of degree less than or equal to $j$. By Lemma 5, $F^{(n)}(x) \to F(x)$ pointwise on $[0, 1]$.

We next strengthen this pointwise convergence to uniform convergence. Since $\mathcal{P}_{j-1}$ is a finite-dimensional vector space, all norms on $\mathcal{P}_{j-1}$ are equivalent. In particular, point evaluations at $j$ distinct points determine a polynomial in $\mathcal{P}_{j-1}$. Choose distinct nodes $x_1, \ldots, x_j \in [0, 1]$, and let $L_1, \ldots, L_j$ be the associated Lagrange basis of $\mathcal{P}_{j-1}$. Then

$$F_j^{(n)}(x) = \sum_{r=1}^{j} F_j^{(n)}(x_r) L_r(x), \qquad F_j(x) = \sum_{r=1}^{j} F_j(x_r) L_r(x).$$

Since $F_j^{(n)}(x_r) \to F_j(x_r)$ for each $r$, the coefficients in this basis converge, and therefore

$$\|F_j^{(n)} - F_j\|_{\sup([0,1])} \to 0.$$

Now split the $j$-th component of the quadrature formula into its interior and endpoint parts:

$$(c^n)_j = \frac{1}{n}\sum_{l=1}^{n-1} F_j^{(n)}\left(\frac{l}{n}\right) f(lh) + \frac{1}{n}\left((\alpha_0^{(n)})_j f(0) + (\alpha_n^{(n)})_j f(T)\right).$$

By the explicit coefficient formulas in Lemma 3 (see also Appendix D), the endpoint coefficients are uniformly bounded:

$$|(\alpha_0^{(n)})_j| + |(\alpha_n^{(n)})_j| \le C_j \qquad \text{for all } n$$

for some constant $C_j > 0$ depending only on $j$ and the discretization method. Since $f$ is Riemann integrable on $[0,T]$, it is bounded, and therefore the endpoint contribution is $O(1/n)$.

Now it suffices to show that the $j$-th component of the limit function $F(x)\colon [0,1] \to \mathbb{R}^N$ is equal to $\sqrt{2j-1}$ times the $(j-1)$-th shifted Legendre polynomial, $\sqrt{2j-1}P_{j-1}(2x-1)$. Once this is shown, we can apply the result of Corollary 2 to conclude the proof. Recall that the exact form of $F$ is:

$$F(x) = V\,\mathrm{diag}\left(1, 2x, 3x^2, \ldots, jx^{j-1}, \ldots, Nx^{N-1}\right) V^{-1}e_1.$$

Differentiating both sides with respect to $x$, we get

$$F'(x) = V\,\mathrm{diag}\left(0, 2, 6x, \ldots, j(j-1)x^{j-2}, \ldots, N(N-1)x^{N-2}\right) V^{-1}e_1.$$

Combining these two equations, we obtain the following differential equation that holds for all $x \in [0,1]$:

$$xF'(x) = (A - I)F(x). \tag{14}$$

Since we know the exact form of $A$, we can derive a recurrence relation for $F$ for arbitrary dimension $N$. Rewriting the $j$-th component $F$ as $F_j(x) = \sqrt{2j-1}f_{j-1}(x)$, we obtain the recurrence relation

$$xf_j'(x) = jf_j(x) + \sum_{l=0}^{j-1}(2l+1)f_l(x). \tag{15}$$

Notice that equation 15 is exactly the recurrence relation satisfied by the $j$-th shifted Legendre polynomial $\tilde{P}_j$. Matching the initial condition $F(1) = Ae_1 = B$, we have $f_j(1) = 1$ for all $j \in \{1, 2, \ldots, N\}$. Then by induction, we can prove that $f_j(x) = \tilde{P}_j(x) = P_j(2x-1)$. Finally, utilizing the uniqueness of the solution for the IVP defined with ODE equation 14 and initial condition at $x = 1$, we arrive at the conclusion:

$$F_j(x) = \sqrt{2j-1}P_{j-1}(2x-1).$$

$\square$

## 5 Convergence rate analysis

In this section, we analyze the convergence rates of the numerical solutions obtained by the discretization methods introduced in Section 2.

**Theorem 3** (Convergence rates of numerical solutions). *Consider the setup in Theorem 2. Suppose in addition that $f$ has bounded variation on $[0,T]$. Then we obtain an $\mathcal{O}(1/n)$ convergence rate for all the forward Euler, backward Euler, bilinear, approximate bilinear, and zero-order hold methods defined in Section 2. If we further assume that $f \in C^2([0,T])$, then the bilinear method achieves an $\mathcal{O}(1/n^2)$ convergence rate.*

*Proof.* Define $T(n) = \frac{1}{n} \sum_{l=0}^{n} P_{j-1} \left( \frac{2l}{n} - 1 \right) f(lh) - \frac{1}{2n} \left( P_{j-1}(-1)f(0) + P_{j-1}(1)f(nh) \right)$ as the result of applying the composite trapezoidal rule to the exact solution. Then we can construct a triangle inequality as

$$\left| (c^n)_j - \frac{\sqrt{2j-1}}{t} \int_0^t P_{j-1} \left( \frac{2s}{t} - 1 \right) f(s) ds \right|$$

$$\leq \underbrace{\left| (c^n)_j - \sqrt{2j-1} T(n) \right|}_{(1)} + \underbrace{\sqrt{2j-1} \left| T(n) - \frac{1}{t} \int_0^t P_{j-1} \left( \frac{2s}{t} - 1 \right) f(s) ds \right|}_{(2)}. \tag{16}$$

(1) corresponds to the error between the numerical solution and the Riemann sum (obtained by applying the trapezoidal rule). For (2), we note that for fixed $t > 0$, $v(x) = P_{j-1}(\frac{2x}{t})f(x)$ is of bounded variation, and bounded by $V > 0$ in closed interval $[0, t]$. Denoting $h = t/n$ and $V_a^b(v)$ to be the total variation of $v$ on the interval $[a, b]$, we obtain

$$\left| T(n) - \frac{1}{t} \int_0^t P_{j-1}(\frac{2s}{t} - 1) f(s) ds \right| \leq \left| \frac{1}{n} \sum_{i=1}^n v(nh) - \frac{1}{nh} \int_0^{nh} v(s) ds \right| + \frac{1}{2n} |v(0) - v(1)|$$

$$\leq \frac{1}{nh} \sum_{i=1}^n \int_{(i-1)h}^{ih} |v(s) - v(ih)| \, ds + \frac{V}{n}$$

$$\leq \frac{1}{n} \sum_{i=1}^n \left( \sup_{x \in [(i-1)h, ih]} v(x) - \inf_{x \in [(i-1)h, ih]} v(x) \right) + \frac{V}{n}$$

$$\leq \frac{1}{n} \sum_{i=1}^n V_{(i-1)h}^{ih}(v) + \frac{V}{n} \leq \frac{2V_0^t(v)}{n}.$$

For the asymptotic rate of (1), recall for $(c^n)_j = \frac{1}{n} \sum_{l=0}^n \left( \alpha_l^{(n)} \right)_j f^l$, the interpolating function $F_j^{(n)}$ was defined such that $F_j^{(n)} \left( \frac{l}{n} \right) = \left( \alpha_l^{(n)} \right)_j$ for $l = \{1, \ldots, n-1\}$ (endpoints are excluded). It was proved in Lemma 5 and Theorem 2 that

$$\lim_{n \to \infty} F^{(n)}(x) = F(x) = V \text{diag} \left( 1, 2x, 3x^2, \ldots, Nx^{N-1} \right) V^{-1} e_1,$$

and that the $j$-th component of $F(x)$ is the scaled-shifted $(j-1)$-th Legendre polynomial, i.e., $F_j(x) = \sqrt{2j-1} P_{j-1}(2x - 1)$. Rewriting (1), we obtain

$$\left| (c^n)_j - \sqrt{2j-1} T(n) \right| = \left| \frac{1}{n} \sum_{l=0}^n \left( \alpha_l^{(n)} \right)_j f^l - \sqrt{2j-1} T(n) \right|$$

$$\leq \frac{1}{n} \sum_{l=1}^{n-1} \left| F_j^{(n)} \left( \frac{l}{n} \right) f^l - \sqrt{2j-1} P_{j-1} \left( \frac{2l}{n} - 1 \right) f^l \right|$$

$$+ \frac{1}{n} \underbrace{\left| \left( \alpha_0^{(n)} \right)_j f^0 + \left( \alpha_n^{(n)} \right)_j f^n - \frac{\sqrt{2j-1}}{2} \left( P_{j-1}(-1)f^0 + P_{j-1}(1)f^n \right) \right|}_{=K_n}$$

$$= \frac{M}{n} \sum_{l=1}^{n-1} \left| F_j^{(n)} \left( \frac{l}{n} \right) - F_j \left( \frac{l}{n} \right) \right| + \frac{K_n}{n}$$

$$\leq M \underbrace{\| F_j^{(n)} - F_j \|_{\sup}}_{(\star)} + \frac{K_n}{n}$$

where $M$ is an upper bound for $f$. Note that $K_n$ is uniformly bounded, so the second part automatically is of $\mathcal{O}(1/n)$. For the first term, note that $(\star)$ differs depending on which discretization method we are

considering. Starting with the forward Euler method, recall that from Lemma 5 that the interpolating function $F_j$ is defined as

$$F_j^{(n)}(x) = e_j^t V \text{diag} \left( \frac{n}{n-1}, \frac{n(nx-1)}{(n-1)(n-2)}, \ldots, \frac{n \prod_{k=1}^{N-1}(nx-k)}{\prod_{k=1}^{N}(n-k)} \right) DV^{-1} e_1.$$

for $x \in [0,1]$. Observing the diagonal components, for every $x \in [0,1]$, we can write the component of $F_j(x) - F_j^{(n)}(x)$ as $\frac{p_x(n)}{q(n)}$ with some polynomials $p_x$, $q$. Note $F_j(x)$ is constant with respect to $n$. Taking closer look at the numerators of $F_j^{(n)}(x)$, we can check the coefficients of $p_x(n)$ are polynomials with respect to $x$. Denote the coefficient of leading term as $a(x)$. Since $\lim_{n\to\infty} \frac{p_x(n)}{q(n)} = 0$, we have $\lim_{n\to\infty} n\frac{p_x(n)}{q(n)} = a(x) \leq \max_{x\in[0,1]} |a(x)|$. As $F_j(x) - F_j^{(n)}(x)$ is finite dimensional, we conclude $\|F_j - F_j^{(n)}\|_{\sup} = \mathcal{O}(1/n)$ for all $j \in \{1, \ldots, N\}$.

The same argument holds with the backward Euler method, approximate bilinear method, and zero-order hold, with the only difference in computing $(\star)$. From Lemma 5, we have:

$$(F_j^{(n)})_{\text{backward}}(x) = n e_j^t V \text{diag} \left( \frac{1}{n+1}, \frac{(nx+1)}{(n+1)(n+2)}, \ldots, \frac{\prod_{k=1}^{N-1}(nx+k)}{\prod_{k=1}^{N}(n+k)} \right) DV^{-1} e_1$$

$$(F_j^{(n)})_{\text{approx bilin}}(x) = e_j^t V \text{diag} \left( \frac{2n}{2n+1}, \frac{n^2 x}{n(n+1)}, \ldots, \frac{n \prod_{k=1}^{j-1}(nx-j/2+k)}{\prod_{k=1}^{j}(n-j/2+k)} \right) DV^{-1} e_1$$

$$(F_j^{(n)})_{\text{zoh}}(x) = e_j^t V \text{diag} \left( 1, 2x + \frac{1}{n}, \ldots, \frac{1}{n^{N-1}} \sum_{k=0}^{N-1} \binom{N}{k} (nx)^k \right) V^{-1} e_1.$$

Using the same argument, all methods shown above achieve $1/n$ convergence rate.

For the bilinear method, we now assume that $f \in C^2([0,T])$. Starting from equation 16, we first know that (2) is of $\mathcal{O}(1/n^2)$ by classical quadrature results for smooth $f$. For the asymptotic rate of (1), we have to consider the asymptotic rate of both $(\star)$ and $K_n$.

The convergence rate for $(\star)$ could be obtained in a similar manner. From Lemma 5, $F_j^{(n)}$ obtained by applying the bilinear method is

$$F_j^{(n)}(x) = e_j^t V \text{diag} \left( \frac{n^2}{n^2 - 1/4}, \frac{n^2 x}{n^2 - 1}, \ldots, \frac{n^2 \prod_{k=1}^{N-1}(nx - N/2 + k)}{\prod_{k=1}^{N+1}(n - N/2 + k - 1)} \right) DV^{-1} e_1.$$

for $x \in [0,1]$. Observe that denominator of the $j$-th term in the diagonal matrix is calculated as $\prod_{k=1}^{j+1}(n-j/2+k-1) = n \prod_{k=1}^{j/2}(n^2 - k^2)$ when $j$ is even, and $\prod_{k=1}^{j+1}(n-j/2+k-1) = \prod_{k=1}^{(j+1)/2}(n^2 - (k-1/2)^2)$ when $j$ is odd. In both cases, the subleading term (w.r.t. $n$) is 2 orders less than the leading term. Then, as before, writing a component of $F_j(x) - F_j^{(n)}$ as $\frac{p_x(n)}{q(n)}$, we have $\lim_{n\to\infty} n^2 \frac{p_x(n)}{q(n)} = a(x) \leq \max_{x\in[0,1]} |a(x)|$. Hence we conclude $\|F_j - F_j^{(n)}\|_{\sup} = \mathcal{O}(1/n^2)$.

For the second term, note that

$$K_n \leq M \left| \left(\alpha_0^{(n)}\right)_j - \frac{\sqrt{2j-1}}{2} P_{j-1}(-1) \right| + M \left| \left(\alpha_n^{(n)}\right)_j - \frac{\sqrt{2j-1}}{2} P_{j-1}(1) \right|.$$

It suffices to show that each term on the right-hand side is $\mathcal{O}(1/n)$. Also, note that

$$\left( \sqrt{2j-1} P_{j-1}(-1) \right)_j = \left( (-1)^{j-1} \sqrt{2j-1} \right)_j = V \text{diag}(1, 0, \ldots, 0) V^{-1} e_1$$

$$\left( \sqrt{2j-1} P_{j-1}(1) \right)_j = \left( \sqrt{2j-1} \right)_j = V \text{diag}(1, 2, \ldots, N) V^{-1} e_1.$$

Recalling that $\tilde{Q}_n = \prod_{j=1}^n \left(I - \frac{1}{2j}A\right)$ and $\tilde{R}_n = \prod_{j=1}^n \left(I + \frac{1}{2j}A\right)^{-1}$, the exact expression for the numerical solution obtained by the bilinear method is

$$c^n = \tilde{Q}_{n-1}\tilde{R}_n \left(I + A/2\right) c^1 + \tilde{Q}_{n-1}\tilde{R}_n \sum_{l=1}^{n-1} \tilde{Q}_l^{-1}\tilde{R}_l^{-1} \frac{1}{2} \left(\frac{1}{l}Bf^l + \frac{1}{l+1}Bf^{l+1}\right).$$

Now, consider the rightmost endpoint, i.e. the coefficient of $f^n$. From the expression above, we can immediately find

$$\left(\alpha_n^{(n)}\right)_j = \frac{1}{2}\left(I + \frac{1}{2n}A\right)^{-1} Bf^n = \frac{1}{2}V\left(I + \frac{1}{2n}D\right)^{-1} V^{-1}e_1 f^n$$

$$= \frac{1}{2}V \text{diag}\left(\frac{1}{1+1/2n}, \frac{2}{1+1/n}, \dots, \frac{N}{1+N/2n}\right) V^{-1}e_1 f^n.$$

From this expression, we can directly verify that $\left|\left(\alpha_n^{(n)}\right)_j - \frac{\sqrt{2j-1}}{2}P_{j-1}(1)\right| = \mathcal{O}(1/n)$ for all $j \in \{1, 2, \dots, N\}$. It remains to check the leftmost endpoint, i.e. coefficient of $f^0$. The terms containing $t = 0$ in $c^n$ are

$$\tilde{Q}_{n-1}\tilde{R}_n \left(I + A/2\right) c^1 = \tilde{Q}_{n-1}\tilde{R}_n \left(c^0 + \frac{h}{2}(I+A)^{-1} Bf'(0) + \frac{B}{2}f^1\right).$$

Notice that due to the extra $h = \mathcal{O}(1/n)$ term, $f'(0)$ term is negligible. Now considering the remaining term for the coefficient of $f^0$,

$$\left(\alpha_0^{(n)}\right)_j = n\tilde{Q}_{n-1}\tilde{R}_n c^0 = nV\left(I + \frac{1}{2n}D\right)^{-1} \prod_{k=1}^{n-1}\left(I - \frac{1}{2k}A\right)\left(I + \frac{1}{2k}A\right)^{-1} V^{-1}e_1 f^0$$

$$= nV\left(I + \frac{1}{2n}D\right)^{-1} \left(\text{diag}\left(\prod_{k=1}^{n-1}\left(\frac{k-1/2}{k+1/2}\right), \dots, \prod_{k=1}^{n-1}\left(\frac{k-N/2}{k+N/2}\right)\right)\right) V^{-1}e_1 f^0$$

$$= V\left(I + \frac{1}{2n}D\right)^{-1} \left(\text{diag}\left(\frac{n/2}{n-1/2}, 0, \dots, \prod_{k=1}^{n-1}\left(\frac{k-N/2}{k+N/2}\right)\right)\right) V^{-1}e_1 f^0.$$

Note that the prefix term $\left(I + \frac{1}{2n}D\right)^{-1}$ does not affect the asymptotic rate with respect to $n$. Observe that the $j$-th term in the diagonal matrix is 0 if $j$ is an even number, and $\mathcal{O}(1/n^{j-1})$ if $j$ is an odd number. Hence $\left|\left(\alpha_0^{(n)}\right)_j - \frac{\sqrt{2j-1}}{2}P_{j-1}(-1)\right| = \mathcal{O}(1/n)$ for all $j \in \{1, 2, \dots, N\}$. $\qquad \square$

**Remark 5.1.** *As discussed earlier, the classical technique of bounding the global error of ODEs by adding up the LTEs is not applicable due to the non-regularity of $f$, and the singularity at $t = 0$. On the other hand, the quadrature formulation only requires $f$ to have bounded variation for $\mathcal{O}(1/n)$ rate, which does not impose strong local conditions on the input function $f$.*

**Remark 5.2.** *The derived convergence rates are tight in the sense that when certain polynomials are used as the input function, the global error matches the upper bound. Here, we denote a bound by $\Theta(g(n))$ if the global error is bounded from both below and above by $c_1 g(n)$ and $c_2 g(n)$, for some positive constants $c_1$, $c_2$, as $n \to \infty$. By direct computation, one can show that the input function $f(t) = t^2$ yields a global error of $\Theta(1/n)$ for all methods except the bilinear method. (The bilinear method yields the exact solution in this case.) Similarly, for the bilinear method, input function $f(t) = t^3$ attains a global error of $\Theta(1/n^2)$. For other methods, using $f(t) = t^3$ does not improve their convergence rate, and the error remains $\Theta(1/n)$. Notably, our rate analysis and these matching examples show that the approximate bilinear method is genuinely a first-order method while the bilinear method is a second-order method, when applied to the LegS ODE.*

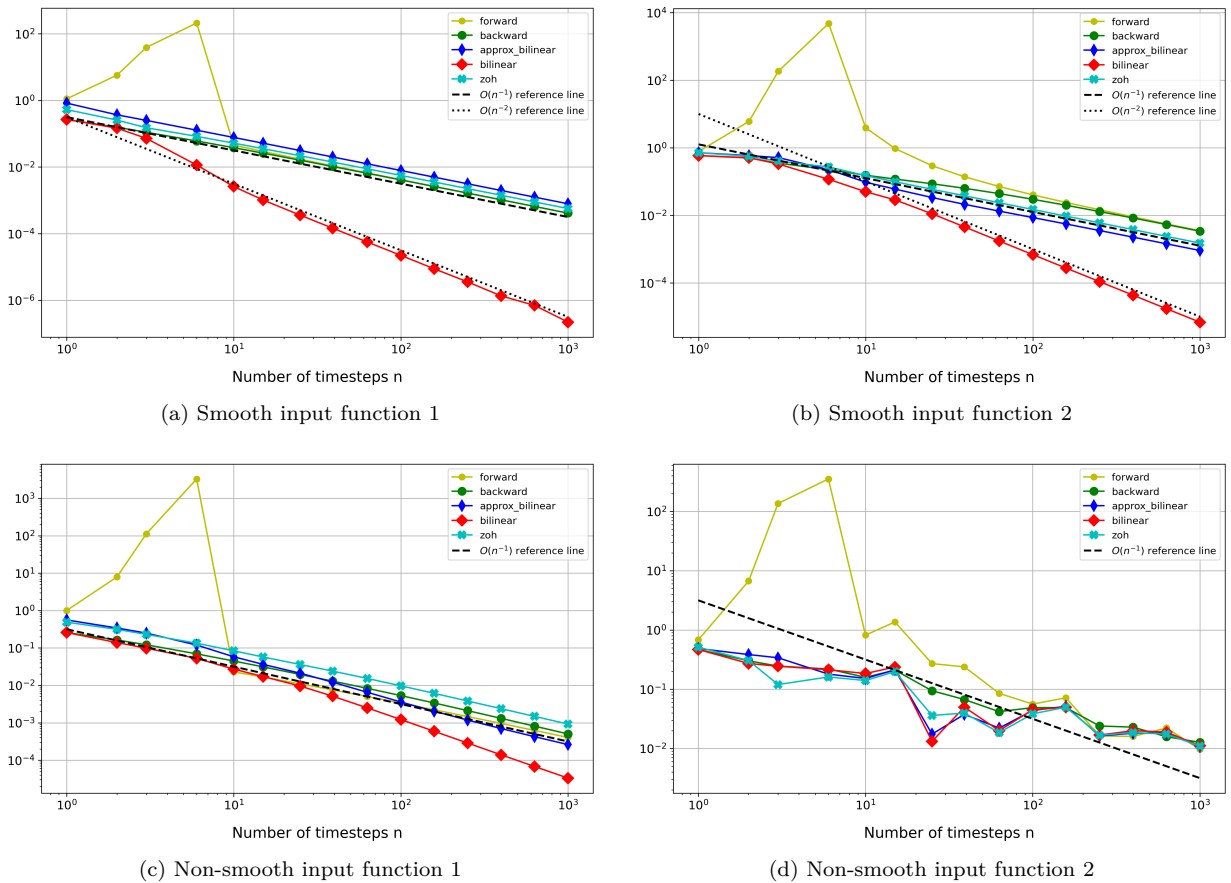

Figure 1: Numerical convergence behavior of the global error of the discretization methods under various regularity properties. The numerical rates agree with theoretical estimates of Theorem 3. **(a)** $f(t) = 2t^3 e^{-t}$ (smooth). Bilinear exhibits $\mathcal{O}(1/n^2)$ rate while others exhibit $\mathcal{O}(1/n)$ rate. **(b)** $f(t) = \frac{1}{4}\sin(10t) + \frac{1}{2}\sin(\frac{10t}{3}) + \sin(\frac{10t}{7})$ (smooth). The qualitative behavior is the same as in (a). **(c)** $f(t) = \sqrt{t}$ (non-smooth, bounded variation). All methods exhibit $\mathcal{O}(1/n)$ rate. **(d)** $f(t) = t^{\frac{1}{20}}\sin(\frac{1}{t})$ (not bounded variation, Riemann integrable). All methods converge in accordance with Theorem 2, but the rates are slower than $\mathcal{O}(1/n)$.

Table 2: Empirical fits of the endpoint global error $E_n = \|c^n - c(t_n)\|$ to the model $E_n \approx Cn^{-p}$ for the discretization methods and test functions of Figure 1. The fitted exponent $\hat{p}$ summarizes the observed slope, while $\hat{C}$ captures the finite-$n$ error magnitude.

| function | method | $\hat{p}$ | $\hat{C}$ |
|---|---|---|---|
| smooth1 | approx_bilinear | 0.999 | 0.785 |
| smooth1 | backward | 0.997 | 0.405 |
| smooth1 | bilinear | 1.962 | 0.184 |
| smooth1 | forward | 1.003 | 0.424 |
| smooth1 | zoh | 0.996 | 0.552 |
| smooth2 | approx_bilinear | 0.976 | 0.773 |
| smooth2 | backward | 0.946 | 2.371 |
| smooth2 | bilinear | 2.000 | 6.848 |
| smooth2 | forward | 1.061 | 5.213 |
| smooth2 | zoh | 0.997 | 1.471 |
| non-smooth1 | approx_bilinear | 1.137 | 0.651 |
| non-smooth1 | backward | 1.027 | 0.609 |
| non-smooth1 | bilinear | 1.566 | 1.641 |
| non-smooth1 | forward | 0.919 | 0.233 |
| non-smooth1 | zoh | 1.023 | 1.085 |
| non-smooth2 | approx_bilinear | 0.597 | 0.735 |
| non-smooth2 | backward | 0.617 | 0.909 |
| non-smooth2 | bilinear | 0.578 | 0.678 |
| non-smooth2 | forward | 0.721 | 1.646 |
| non-smooth2 | zoh | 0.556 | 0.567 |

## 6 Numerical experiments

We perform numerical experiments to complement the theoretical results of Sections 4 and 5. The first experiment studies the asymptotic convergence behavior predicted by Theorems 2 and 3, while the second examines finite-$n$ error constants.

For the convergence-rate experiment, we use dimension $N = 8$, terminal time $T = 2$, and uniform step size $h = 2/n$. We compare the forward Euler, backward Euler, zero-order hold, bilinear, and approximate bilinear methods on the four test inputs shown in Figure 1. The vertical axis reports the endpoint global error $\|c^n - c(t_n)\|$. As Figure 1 shows, the empirical slopes agree with the theory. For the two smooth inputs, the bilinear method exhibits second-order convergence, whereas the other four methods exhibit first-order convergence. For the bounded-variation but non-smooth input $f(t) = \sqrt{t}$, all methods exhibit first-order behavior. Finally, for the oscillatory input $f(t) = t^{1/20}\sin(1/t)$, which is Riemann integrable but not of bounded variation, all methods still converge, but the rate is visibly slower than $O(1/n)$, in agreement with Theorem 2.

To make the finite-resolution behavior more explicit, we also fit the empirical error curves by the model

$$E_n \approx Cn^{-p},$$

where $E_n = \|c^n - c(t_n)\|$, $p$ is the empirical order, and $C$ is the associated error constant. Table 2 reports the fitted parameters for all test functions and discretization methods.

## 7 Conclusion

In this work, we laid the mathematical foundations of the HiPPO-LegS ODE and its discretization methods. First, we established the existence and uniqueness of the solution to the HiPPO-LegS ODE, despite the singularity at $t = 0$, and presented this result in Theorem 1. Next, we provided a framework for analyzing various discretization methods for the LegS ODE by reinterpreting them as numerical quadrature rules.

Using this viewpoint, we proved convergence of various discretization methods for all Riemann integrable input functions and stated the results in Theorem 2. Finally, under additional regularity assumptions, we obtained optimal convergence rates; see Theorem 3.

A particularly promising direction for future work is to extend our theory to higher-order methods—such as Runge–Kutta schemes—or to linear multistep methods applied to the HiPPO-LegS ODE. If convergence guarantees remain valid for non-smooth inputs, possibly with better convergence rates, it could motivate novel HiPPO algorithms.

**Acknowledgments**

EKR was supported as a fellow of the Alfred P. Sloan Foundation.

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

## A    Proof of Lemma 1

*Proof.* We first show $f$ satisfies the desired conditions. Since $f$ is continuous on $(0, 1/2]$, it suffices to show $f$ is continuous at $t = 0$. Calculating the differentiation, we see that

$$f(t) = \frac{d}{dt}\left(\frac{t^2}{\log(1/t)}\sin(1/t)\right) = \frac{2t}{\log(1/t)}\sin(1/t) + \frac{t}{\log^2(1/t)}\sin(1/t) - \frac{1}{\log(1/t)}\cos(1/t)$$

holds for $t \in (0, 1/2]$. Since $\lim_{t\to 0^+}\log(1/t) = \lim_{u\to\infty}\log(u) = \infty$, $\lim_{t\to 0^+}f(t) = 0$. Therefore, $f$ is continuous at $t = 0$, so it is integrable and has a Lebesgue point at 0.

Now, we show that $c(t)$ is not absolutely continuous on $[0, T]$. It is sufficient to prove that the first component $c_1(t)$ is not absolutely continuous on $[0, T]$. Plugging the definition of $f(t)$ to equation 4, we obtain

$$c_1(t) = \begin{cases} c_1\left(\frac{1}{2}\right) & \text{if } t > 1/2 \\ d_1\frac{t}{\log(1/t)}\sin(1/t) & \text{if } t \in (0, 1/2] \\ 0 & \text{if } t = 0. \end{cases}$$

Let $\delta > 0$ be an arbitrary positive number. For $N > 0$ such that $\frac{1}{(2N+1)\pi + \frac{\pi}{2}} < \min\left\{\delta, \frac{1}{2}, T\right\}$, consider

$$t_n = \begin{cases} \left((2N + n + \frac{1}{2})\pi\right)^{-1} & \text{if } n \text{ is odd} \\ \left((2N + n)\pi\right)^{-1} & \text{if } n \text{ is even.} \end{cases}$$

Define $x_n = t_n$ and $y_n = t_{n+1}$ for $n \geq 1$. Then $\sum_{n=1}^{\infty}(y_n - x_n) = t_1 = \frac{1}{(2N+1)\pi + \frac{\pi}{2}} < \delta$. However, since

$$|c_1(x_n) - c_1(y_n)| = |c_1(t_n) - c_1(t_{n+1})| = \begin{cases} |c_1(t_n)| & \text{if } n \text{ is odd} \\ |c_1(t_{n+1})| & \text{if } n \text{ is even,} \end{cases}$$

we have

$$\sum_{n=1}^{\infty}|c_1(x_n) - c_1(y_n)| = |c_1(t_1)| + 2\sum_{m=1}^{\infty}|c_1(t_{2m+1})|$$

$$> 2d_1\sum_{m=1}^{\infty}\frac{t_{2m+1}}{\log(1/t_{2m+1})} = 2d_1\sum_{m=1}^{\infty}\frac{1}{(2N + 2m + 3/2)\pi\log((2N + 2m + 3/2)\pi)}$$

$$> 2d_1\sum_{m=1}^{\infty}\frac{1}{2\pi(N + m + 1)\log(2\pi(N + m + 1))} = 2d_1\sum_{m=N+2}^{\infty}\frac{1}{2m\pi\log(2m\pi)} = \infty,$$

where the last equality follows from integral test. Since $(x_n, y_n) \subset [0, T]$ are disjoint and $\delta > 0$ was arbitrary, we conclude $c_1$ is not absolutely continuous on $[0, T]$. □

## B    Proof of Corollary 1

*Proof.* If we assume that the solution exists, we know by the previous theorem that $c(0) = A^{-1}Bf(0)$. Further, in the proof of Theorem 1, we have that $c(t) = V\tilde{c}(t)$ for all $t \in (0, T]$ where

$$\tilde{c}_j(t) = \frac{\left(V^{-1}B\right)_j}{t^j}\int_0^t s^{j-1}f(s)ds.$$

The $j$-th component of $c$ could be expressed as

$$c_j(t) = \sum_{k=1}^{N}V_{jk}\frac{\left(V^{-1}B\right)_k}{t^k}\int_0^t s^{k-1}f(s)ds = \frac{1}{t}\int_0^t\underbrace{\sum_{k=1}^{N}V_{jk}\left(V^{-1}B\right)_k x^{k-1}}_{=G_j(x)}f(s)ds \qquad (17)$$

where we made the substitution $x = \frac{s}{t}$. Since other terms do not depend on the index $j$, we simplify this expression by analyzing $G_j(x)$, regarding $x$ as a symbolic (differentiable) variable. Expression $G_j(x)$ could be rewritten as

$$G_j(x) = \sum_{k,l=1}^{N} V_{jk} x^{k-1} V_{kl}^{-1} B_l = e_j^t V \operatorname{diag}\left(1, x, \ldots, x^{j-1}, \ldots, x^{N-1}\right) V^{-1} B.$$

Define $G(x)$ as a length $N$ vector with its $j$-th component being $G_j(x)$, which is a polynomial of $x$. Then we obtain a matrix differential equation with respect to the symbolic variable $x$ :

$$x\frac{dG}{dx}(x) = (A - I) G(x).$$

Since we know the precise structure of matrix $A$, letting $g_{j-1}(x) = \frac{G_j(x)}{\sqrt{2j-1}}$, we can obtain the following recurrence relation for $g_j$'s,

$$xg'_{j-1}(x) = (j-1)g_{j-1} + \sum_{k=0}^{j-2}(2k+1)g_k(x),$$

which is precisely the recurrence relation equation 2 for the shifted Legendre polynomials. Since $G_j(1) = B_j = \sqrt{2j-1}$, we conclude that $g_j$ is equal to the $(j-1)$-th shifted Legendre polynomial $\tilde{P}_{j-1}$. Finally, incorporating this observation into equation 17, we can rewrite the solution of the LegS ODE for $t \in (0, T]$ as

$$c_j(t) = \frac{1}{t} \int_0^t \sqrt{2j-1}\tilde{P}_{j-1}(x) f(s)ds = \frac{\sqrt{2j-1}}{t} \int_0^t P_{j-1}\left(\frac{2s}{t} - 1\right) f(s)ds.$$

$\square$

## C   Proof of Lemma 3

*Proof.* For notational simplicity, we define $Q_n$ for $n \geq 2$, and $\tilde{Q}_n, R_n, \tilde{R}_n$ for $n \geq 1$ as follows:

$$Q_n := \prod_{j=1}^{n-1} \left(I - \frac{1}{j+1}A\right), \qquad \tilde{Q}_n := \prod_{j=1}^{n} \left(I - \frac{1}{2j}A\right),$$

$$R_n := \prod_{j=1}^{n} \left(I + \frac{1}{j}A\right)^{-1}, \qquad \tilde{R}_n := \prod_{j=1}^{n} \left(I + \frac{1}{2j}A\right)^{-1}.$$

We use the $\prod$ notation when the multiplications are commutative. Note that all $Q_n, \tilde{Q}_n, R_n, \tilde{R}_n$ are invertible.

We start by proving for the forward Euler method. Recall that the forward Euler method yields the following recurrence relation at step $n$:

$$c^{n+1} = \left(I - \frac{1}{n}A\right) c^n + \frac{1}{n}Bf^n.$$

Repeating this procedure, we can obtain an exact formula for the numerical solution obtained by applying forward Euler method to the LegS ODE. By induction we obtain,

$$\begin{aligned}
c^{n+1} &= \left(I - \frac{1}{n}A\right) c^n + \frac{1}{n}Bf^n \\
&= \left(I - \frac{1}{n}A\right)\left(I - \frac{1}{n-1}A\right) c^{n-1} + \left(I - \frac{1}{n}A\right)\frac{1}{n-1}Bf^{n-1} + \frac{1}{n}Bf^n \\
&= Q_n\left(c^1 + Bf^1\right) + Q_n \sum_{l=2}^{n} \frac{1}{l}Q_l^{-1}Bf^l.
\end{aligned}$$

As explained in Section 2, we 'zero out' the ill-defined iteration, thereby letting $c^1 = c^0$. Hence we have

$$c^n = \frac{1}{n}\sum_{l=0}^{n}\alpha_l^{(n)}f(lh) = Q_{n-1}(e_1f^0 + Bf^1) + Q_{n-1}\sum_{l=2}^{n-1}\frac{1}{l}Q_l^{-1}Bf^l. \tag{18}$$

One can verify that $\alpha_l^{(n)}$ depends only on $l$ and $n$.

For the backward Euler method, we start from

$$c^{n+1} = \left(I + \frac{1}{n+1}A\right)^{-1}c^n + \left(I + \frac{1}{n+1}A\right)^{-1}\frac{1}{n+1}Bf^{n+1}.$$

Then, we can derive inductively

$$
\begin{aligned}
c^{n+1} &= \left(I + \frac{1}{n+1}A\right)^{-1}c^n + \left(I + \frac{1}{n+1}A\right)^{-1}\frac{1}{n+1}Bf^{n+1} \\
&= \left(I + \frac{1}{n+1}A\right)^{-1}\left(I + \frac{1}{n}A\right)^{-1}c^{n-1} + \left(I + \frac{1}{n+1}A\right)^{-1}\left(I + \frac{1}{n}A\right)^{-1}\frac{1}{n}Bf^n \\
&\quad + \left(I + \frac{1}{n+1}A\right)^{-1}\frac{1}{n+1}Bf^{n+1} \\
&= R_{n+1}(c^0 + Bf^1) + R_{n+1}\sum_{l=1}^{n}\frac{1}{l+1}R_l^{-1}Bf^{l+1}.
\end{aligned}
$$

Thus we can verify $c^n$ has the desired form with the specific expression

$$c^n = \frac{1}{n}\sum_{l=0}^{n}\alpha_l^{(n)}f(lh) = R_n(e_1f^0 + Bf^1) + R_n\sum_{l=1}^{n-1}\frac{1}{l+1}R_l^{-1}Bf^{l+1}. \tag{19}$$

For the bilinear method, we start from

$$c^{n+1} = \left(I + \frac{1}{n+1}A/2\right)^{-1}\left(I - \frac{1}{n}A/2\right)c^n + \left(I + \frac{1}{n+1}A/2\right)^{-1}\frac{1}{2}\left(\frac{1}{n}f^n + \frac{1}{n+1}f^{n+1}\right)B.$$

Similarly, by induction, we obtain

$$
\begin{aligned}
c^{n+1} &= \left(I + \frac{1}{n+1}A/2\right)^{-1}\left(I - \frac{1}{n}A/2\right)c^n + \left(I + \frac{1}{n+1}A/2\right)^{-1}\frac{1}{2}\left(\frac{1}{n}Bf^n + \frac{1}{n+1}Bf^{n+1}\right) \\
&= \left(I + \frac{1}{n+1}A/2\right)^{-1}\left(I + \frac{1}{n}A/2\right)^{-1}\left(I - \frac{1}{n}A/2\right)\left(I - \frac{1}{n-1}A/2\right)c^{n-1} \\
&\quad + \left(I + \frac{1}{n+1}A/2\right)^{-1}\left(I + \frac{1}{n}A/2\right)^{-1}\left(I - \frac{1}{n-1}A/2\right)\frac{1}{2}\left(\frac{1}{n-1}Bf^{n-1} + \frac{1}{n}Bf^n\right) \\
&\quad + \left(I + \frac{1}{n+1}A/2\right)^{-1}\frac{1}{2}\left(\frac{1}{n}Bf^n + \frac{1}{n+1}Bf^{n+1}\right) \\
&= \tilde{Q}_n\tilde{R}_{n+1}\left(I + A/2\right)c^1 + \tilde{Q}_n\tilde{R}_{n+1}\sum_{l=1}^{n}\tilde{Q}_l^{-1}\tilde{R}_l^{-1}\frac{1}{2}\left(\frac{1}{l}Bf^l + \frac{1}{l+1}Bf^{l+1}\right).
\end{aligned}
$$

As for the forward Euler case, we 'zero out' the ill-defined term in the first iteration. This yields $c^1 = \left(I + \frac{A}{2}\right)^{-1}c^0 + \left(I + \frac{A}{2}\right)^{-1}\left(\frac{1}{2}f^1\right)$. Then,

$$c^n = \tilde{Q}_{n-1}\tilde{R}_n\left(e_1f^0 + \frac{f^1}{2}\right) + \tilde{Q}_{n-1}\tilde{R}_n\sum_{l=1}^{n-1}\tilde{Q}_l^{-1}\tilde{R}_l^{-1}\frac{1}{2}\left(\frac{1}{l}Bf^l + \frac{1}{l+1}Bf^{l+1}\right).$$

Rearranging terms,

$$c^n = \frac{1}{n} \sum_{l=0}^{n} \alpha_l^{(n)} f(lh)$$

$$= \tilde{Q}_{n-1} \tilde{R}_n e_1 f^0 + \tilde{Q}_{n-1} \tilde{R}_n \sum_{l=1}^{n-1} \frac{1}{2l} \left( \tilde{Q}_l^{-1} \tilde{R}_l^{-1} + \tilde{Q}_{l-1}^{-1} \tilde{R}_{l-1}^{-1} \right) B f^l + \tilde{R}_n \tilde{R}_{n-1}^{-1} \frac{1}{2n} B f^n \tag{20}$$

where we define $\tilde{Q}_0 = \tilde{R}_0 = I$. Thus we recover the desired form for $c^n$.

For the approximate bilinear method, we start from

$$c^{n+1} = \left( I + \frac{1}{n+1} A/2 \right)^{-1} \left( I - \frac{1}{n+1} A/2 \right) c^n + \left( I + \frac{1}{n+1} A/2 \right)^{-1} \left( \frac{1}{n+1} f^{n+1} \right) B.$$

By induction, we obtain

$$c^{n+1} = \left( I + \frac{1}{n+1} A/2 \right)^{-1} \left( I - \frac{1}{n+1} A/2 \right) c^n + \left( I + \frac{1}{n+1} A/2 \right)^{-1} \left( \frac{1}{n+1} B f^{n+1} \right)$$

$$= \left( I + \frac{1}{n+1} A/2 \right)^{-1} \left( I + \frac{1}{n} A/2 \right)^{-1} \left( I - \frac{1}{n+1} A/2 \right) \left( I - \frac{1}{n} A/2 \right) c^{n-1}$$

$$+ \left( I + \frac{1}{n+1} A/2 \right)^{-1} \left( I + \frac{1}{n} A/2 \right)^{-1} \left( I - \frac{1}{n+1} A/2 \right) \left( \frac{1}{n} B f^n \right)$$

$$+ \left( I + \frac{1}{n+1} A/2 \right)^{-1} \left( \frac{1}{n+1} B f^{n+1} \right)$$

$$= \tilde{Q}_{n+1} \tilde{R}_{n+1} c^0 + \tilde{Q}_{n+1} \tilde{R}_{n+1} \sum_{l=1}^{n} \tilde{Q}_{l+1}^{-1} \tilde{R}_l^{-1} \left( \frac{1}{l+1} B f^{l+1} \right).$$

Thus we can verify $c^n$ has the desired form with the specific expression

$$c^n = \tilde{Q}_n \tilde{R}_n c^0 + \tilde{Q}_n \tilde{R}_n \sum_{l=1}^{n-1} \tilde{Q}_{l+1}^{-1} \tilde{R}_l^{-1} \left( \frac{1}{l+1} B f^{l+1} \right). \tag{21}$$

For the Zero-order hold method, we start from

$$c^{n+1} = e^{A \log\left(\frac{n}{n+1}\right)} c^n + \left( I - e^{A \log\left(\frac{n}{n+1}\right)} \right) A^{-1} B f^n.$$

By induction, we obtain

$$c^{n+1} = e^{A \log\left(\frac{n}{n+1}\right)} e^{A \log\left(\frac{n-1}{n}\right)} c^{n-1} + \left( I - e^{A \log\left(\frac{n}{n+1}\right)} \right) A^{-1} B f^n$$

$$+ e^{A \log\left(\frac{n}{n+1}\right)} \left( I - e^{A \log\left(\frac{n-1}{n}\right)} \right) A^{-1} B f^{n-1}$$

$$= e^{A \log\left(\frac{n-1}{n+1}\right)} c^{n-1} + \left( I - e^{A \log\left(\frac{n}{n+1}\right)} \right) A^{-1} B f^n + \left( e^{A \log\left(\frac{n}{n+1}\right)} - e^{A \log\left(\frac{n-1}{n+1}\right)} \right) A^{-1} B f^{n-1}$$

$$= \sum_{k=0}^{n} \left( e^{A \log\left(\frac{n+1-k}{n+1}\right)} - e^{A \log\left(\frac{n-k}{n+1}\right)} \right) A^{-1} B f^{n-k}.$$

Note that in this expression, we are denoting (with abuse of notation) $e^{\log 0} = 0$. Thus we can verify $c^n$ has the desired form with the specific expression

$$c^n = \frac{1}{n} \sum_{l=0}^{n} \alpha_l^{(n)} f(lh) = \sum_{l=0}^{n-1} \left( e^{A \log\left(\frac{l+1}{n}\right)} - e^{A \log\left(\frac{l}{n}\right)} \right) A^{-1} B f^l. \tag{22}$$

$\square$

## D    Remaining Proof of Lemma 5

In this section, we provide the proof of Lemma 5 for the other methods, i.e., backward Euler, bilinear, approximate bilinear, and zero-order hold.

*Proof.* For the backward Euler method, referring to equation 19, the interpolating function satisfies

$$p^{(n)}\left(\frac{l}{n}\right) = \frac{n}{l}\prod_{k=l}^{n}\left(I+\frac{1}{k}A\right)^{-1}Ae_1$$

for $l \in \{1, 2, \ldots, n-1\}$. Then the product term in the RHS is equal to

$$\prod_{k=l}^{n}\left(I+\frac{1}{k}A\right)^{-1} = \prod_{k=l+1}^{n}V\left(I+\frac{1}{k}D\right)^{-1}V^{-1} = V\left(\prod_{k=l}^{n}\left(I+\frac{1}{k}D\right)\right)^{-1}V^{-1}$$

$$= V\left(\operatorname{diag}\left(\prod_{k=l}^{n}\left(1+\frac{1}{k}\right)^{-1},\prod_{k=l}^{n}\left(1+\frac{2}{k}\right)^{-1},\ldots,\prod_{k=l}^{n}\left(1+\frac{j}{k}\right)^{-1}\right)\right)V^{-1}$$

and by canceling out terms assuming $n$ is large,

$$\frac{1}{l}\prod_{k=l}^{n}\left(1+\frac{i}{k}\right)^{-1} = \frac{1}{l}\frac{\prod_{k=l}^{n}k}{\prod_{k=l}^{n}(k+i)} = \frac{\prod_{k=1}^{i-1}(l+k)}{\prod_{k=1}^{i}(n+k)}.$$

Similar to the forward Euler case, we can define the interpolating polynomial $F_j^{(n)}$ as

$$F_j^{(n)}(x) = ne_j^tV\operatorname{diag}\left(\frac{1}{n+1},\frac{(nx+1)}{(n+1)(n+2)},\ldots,\frac{\prod_{k=1}^{N-2}(nx+k)}{\prod_{k=1}^{N-1}(n+k)},\frac{\prod_{k=1}^{N-1}(nx+k)}{\prod_{k=1}^{N}(n+k)}\right)DV^{-1}e_1.$$

By the same reasoning as for the forward Euler case, we conclude that $F_j^{(n)}$ is a $(j-1)$-degree polynomial. Moreover, it converges to equation 13 pointwise for $x \in [0,1]$ as $n \to \infty$.

For the bilinear method, referring to equation 20, the interpolating function satisfies

$$p^{(n)}\left(\frac{l}{n}\right) = \frac{n}{2l}\left(I+\frac{1}{2n}A\right)^{-1}\times$$

$$\left(\prod_{k=l}^{n-1}\left(I-\frac{1}{2k}A\right)\left(I+\frac{1}{2k}A\right)^{-1}+\prod_{k=l+1}^{n-1}\left(I-\frac{1}{2k}A\right)\left(I+\frac{1}{2k}A\right)^{-1}\right)Ae_1$$

$$= \frac{n}{2l}\left(I+\frac{1}{2n}A\right)^{-1}\times$$

$$\left(I+\left(I-\frac{1}{2l}A\right)\left(I+\frac{1}{2l}A\right)^{-1}\right)\prod_{k=l+1}^{n-1}\left(I-\frac{1}{2k}A\right)\left(I+\frac{1}{2k}A\right)^{-1}Ae_1$$

for $l \in \{1, 2, \ldots, n-1\}$. For the RHS, the last term simplifies to

$$\prod_{k=l+1}^{n-1}\left(I-\frac{1}{2k}A\right)\left(I+\frac{1}{2k}A\right)^{-1}$$

$$= V\prod_{k=l+1}^{n-1}\left(I-\frac{1}{2k}D\right)\left(I+\frac{1}{2k}D\right)^{-1}V^{-1}$$

$$= V\left(\operatorname{diag}\left(\prod_{k=l+1}^{n-1}\left(\frac{k-1/2}{k+1/2}\right),\prod_{k=l+1}^{n-1}\left(\frac{k-1}{k+1}\right),\ldots,\prod_{k=l+1}^{n-1}\left(\frac{k-j/2}{k+j/2}\right)\right)\right)V^{-1}$$

and the prefix terms simplify to

$$\frac{n}{2l}\left(I+\frac{1}{2n}A\right)^{-1}\left(I+\left(I-\frac{1}{2l}A\right)\left(I+\frac{1}{2l}A\right)^{-1}\right)$$

$$=\frac{n}{2l}V\left(I+\frac{1}{2n}D\right)^{-1}\left(I+\left(I-\frac{1}{2l}D\right)\left(I+\frac{1}{2l}D\right)^{-1}\right)V^{-1}$$

$$=V\left(\mathrm{diag}\left(\left(\frac{n^2}{(n+1/2)(l+1/2)}\right),\left(\frac{n^2}{(n+1)(l+1)}\right),\ldots,\left(\frac{n^2}{(n+1/2j)(l+1/2j)}\right)\right)\right)V^{-1}.$$

Combining these two terms and simplifying the denominators and numerators, we obtain

$$p^{(n)}\left(\frac{l}{n}\right)=V\mathrm{diag}\left(\frac{n^2}{n^2-1/4},\ldots,\frac{\prod_{k=1}^{N}(l-N/2+k)}{\prod_{k=1}^{N}(n-N/2+k-1)}\cdot\frac{n^2}{(l+j/2)(n+j/2)}\right)DV^{-1}e_1.$$

Define $F_j^{(n)}$ as

$$F_j^{(n)}(x)=e_j^tV\mathrm{diag}\left(\frac{n^2}{(n-1/2)(n+1/2)},\ldots,\frac{\prod_{k=1}^{N-1}(nx-N/2+k)}{\prod_{k=1}^{N}(n-N/2+k-1)}\cdot\frac{n^2}{n+N/2}\right)DV^{-1}e_1.$$

By the same reasoning as for the forward Euler case, we conclude that $F_j^{(n)}$ is a $(j-1)$-degree polynomial. Moreover, it converges to equation 13 pointwise for $x\in[0,1]$ as $n\to\infty$.

For the approximate bilinear method, referring to equation 21, the interpolating function satisfies

$$p^{(n)}\left(\frac{l}{n}\right)=\frac{n}{l}\prod_{k=l+1}^{n}\left(I-\frac{1}{2k}A\right)\prod_{k=l}^{n}\left(I+\frac{1}{2k}A\right)^{-1}Ae_1$$

$$=\frac{n}{l}\left(I+\frac{1}{2l}A\right)^{-1}\prod_{k=l+1}^{n}\left(I-\frac{1}{2k}A\right)\left(I+\frac{1}{2k}A\right)^{-1}Ae_1$$

for $l\in\{1,2,\ldots,n-1\}$. Assuming $n$ is large, the product term in RHS simplifies as

$$\prod_{k=l+1}^{n}\left(I-\frac{1}{2k}A\right)\left(I+\frac{1}{2k}A\right)^{-1}$$

$$=\prod_{k=l+1}^{n}V\left(I-\frac{1}{2k}D\right)\left(I+\frac{1}{2k}D\right)^{-1}V^{-1}$$

$$=V\left(\mathrm{diag}\left(\prod_{k=l+1}^{n}\left(\frac{k-1/2}{k+1/2}\right),\prod_{k=l+1}^{n}\left(\frac{k-1}{k+1}\right),\ldots,\prod_{k=l+1}^{n}\left(\frac{k-j/2}{k+j/2}\right)\right)\right)V^{-1}$$

and the prefix term simplifies to

$$\frac{1}{l}\left(I+\frac{1}{2l}A\right)^{-1}=\frac{1}{l}V\left(I+\frac{1}{2l}D\right)^{-1}V^{-1}=V\left(\mathrm{diag}\left(\frac{1}{l+1/2},\frac{1}{l+3/2}\ldots,\frac{1}{l+j/2}\right)\right)V^{-1}.$$

Assuming that $n$ is large and canceling out terms, we obtain

$$p^{(n)}\left(\frac{l}{n}\right)=V\mathrm{diag}\left(\frac{2n}{2n+1},\frac{ln}{n(n+1)},\ldots,\frac{\prod_{k=1}^{N}(l-N/2+k)}{\prod_{k=1}^{N}(n-N/2+k)}\cdot\frac{n}{l+N/2}\right)DV^{-1}e_1$$

Define $F_j^{(n)}$ as

$$F_j^{(n)}(x)=e_j^tV\mathrm{diag}\left(\frac{2n}{2n+1},\frac{n^2x}{n(n+1)},\ldots,\frac{n\prod_{k=1}^{j-1}(nx-j/2+k)}{\prod_{k=1}^{j}(n-j/2+k)}\right)DV^{-1}e_1$$

By the same reasoning as in the forward Euler case, we conclude that $F_j^{(n)}$ is a $(j-1)$-degree polynomial. Moreover, it converges to equation 13 pointwise for $x \in [0,1]$ as $n \to \infty$.

For zero-order hold, referring to equation 22, the interpolating function satisfies

$$p^{(n)}\left(\frac{l}{n}\right) = n\left(e^{A\log\left(\frac{l+1}{n}\right)} - e^{A\log\left(\frac{l}{n}\right)}\right)e_1$$

for $l \in \{1, 2, \ldots, n-1\}$. Diagonalize $A$ by $A = VDV^{-1}$ and

$$p^{(n)}\left(\frac{l}{n}\right) = nV\left(e^{D\log\left(\frac{l+1}{n}\right)} - e^{D\log\left(\frac{l}{n}\right)}\right)V^{-1}e_1$$

$$= V\mathrm{diag}\left(1, \ldots, \frac{1}{n^{j-1}}\left((l+1)^j - l^j\right), \ldots, \frac{1}{n^{N-1}}\left((l+1)^N - l^N\right)\right)V^{-1}e_1.$$

As in the other cases, we can define $F_j^{(n)}$ as

$$F_j^{(n)}(x) = e_j^t V\mathrm{diag}\left(1, 2x + \frac{1}{n}, \frac{1}{n^{j-1}}\sum_{k=0}^{j-1}\binom{j}{k}(nx)^k, \ldots, \frac{1}{n^{N-1}}\sum_{k=0}^{N-1}\binom{N}{k}(nx)^k\right)V^{-1}e_1$$

which pointwise converges to equation 13. $\qquad\square$

