# OpenReview forum: "Numerical Analysis of HiPPO-LegS ODE for Deep State Space Models"
_TMLR — Accepted by TMLR_

### Review · Reviewer_AfdB · 2025-12-19

**Summary Of Contributions:**

The paper is a pure theory paper, focusing on the mathematical properties of HiPPO-LegS ODE and the discretization. The paper proves the existence and uniqueness of the solution to the LegS ODE, and convergence of the common discretization schemes to the continuous-time ODE, and the convergence rate.

**Audience:**

Yes

**Audience Explanation:**

I think some individuals in TMLR's audience might be interested in knowing the findings of this paper. But the people who are interested in this paper are probably those who are already familiar with the role of HiPPO-LegS ODE in deep learning because the whole paper focuses on studying the theoretical properties of HiPPO-LegS ODE without any discussing on the implications in deep learning.

**Claims And Evidence:**

Yes

**Claims Explanation:**

This is a pure theory paper, focusing on studying the theoretical properties of HiPPO-LegS ODE. I have no doubt the mathematical part is rigorous. But there are not enough discussions how the theory or theoretical findings in this paper connect to deep learning, and how the results can shed some new insight in machine learning.

**Requested Changes:**

(1) In the introduction section, around equation (1), there should be some discussions how the ODE in equation (1) is used in deep learning. For example, what are the interpretations of $c(t)$, $A$ and $B$ in the context of deep learning. Why the ODE is of the form of equation (1). What are the interpretations of existence, uniqueness and convergence of solution of the ODE in deep learning. My overall impression is that the paper is a purely theoretical paper, studying the properties of equation (1). But there should be some interpretations of the theoretical results in terms of the insights in machine learning.

(2) In the paragraph after equation (1), write $(j-1)$-th instead of $j-1$-th.

(3) After equation (1), the introduction of $c_{j}(t)$ is a bit confusing, and you need to remind the readers what $c_{j}(t)$ has anything to do with $c(t)$.

(4) The numerical experiments are only illustrating the theoretical aspects, without a direct connection to any applications in deep learning. It would be helpful to illustrate HiPPO-LegS ODE in the context of machine learning in numerical experiments.

(5) In the second paragraph in Section 1.1. what is S4 on the third line?

(6) In the paragraph before Section 4.1., there should be space between signals and Gu et al. (2020).

(7) The second sentence in the statement of Theorem 3 seems to have some English problem.

(8) In Theorem 3, you showed that when $f$ is of bounded variation on $[0,T]$, then you have $O(1/n)$ convergence rate for all discretization schemes, and if you further assume $f$ is $C^2$, then you obtain $O(1/n^2)$ convergence rate for the bilinear method. One natural question is that if $f$ is of bounded variation, every method achieves the same convergence rate. Then, in practice, which one should be used. It would be nice if there are some discussions about the advantage and disadvantage applying each method in the context of LegS ODE.

(9) In Remark 5.2, you said that $f(t)=t^2$ yields a global error of order $1/n$ for all methods except the bilinear method,
and for the bilinear method, $f(t)=t^3$ yields a global error of order $1/n^2$. It is a bit confusing you are comparing bilinear method
with other methods when you use two different input functions. Please comment on the global error for bilinear method for $f(t)=t^2$
and the global error for all the other methods for $f(t)=t^3$.

---

> ### Author Response · Authors · 2026-04-21
>
> We thank the reviewer for the careful reading and for the detailed suggestions.
> Below, we address the reviewer's concerns individually.
>
> 1. We agree that this needs to be explained more clearly. In the revision, we added a sentence discussing the meaning and implication of Eq (1), including the role of $c(t)$ in machine learning applications.
>
> 2. We fixed this in the revised text.
>
> 3. We added a line clarifying the meaning of $c_j(t)$ after Eq (1).
>
> 4. We appreciate this suggestion. However, the goal of the numerical section is not to benchmark end-to-end machine-learning performance, but to validate the numerical claims proved in the paper.
> For that purpose, direct convergence experiments on the underlying ODE and its discretization schemes are the most faithful validation, since they isolate exactly the effect analyzed in the theory.
> To strengthen the practical comparison within the scope of the paper, we instead added finite-$n$ error-constant comparisons across schemes in addition to the original rate plots.
>
> 5. S4 is the model suggested by Gu et al. [1], which uses the HiPPO-LegS ODEs for initialization.
>
> 6. We fixed this in the revised text.
>
> 7. We agree that the statement of Theorem 3 was confusing. We have rewritten the statement.
>
> 8. We agree that this is an important point. Since the theoretical analysis reveals that functions of bounded variation (but not necessarily smooth) have the same convergence rate, it would require to analyze the coefficient of convergence to compare between the methods. We have conducted additional experiments verifying this; please refer to the General Response.
>
> 9. We agree that the previous wording was too terse. The point of Remark 5.2 is not to compare all methods on the same test function, but to show tightness of the upper bounds for the different method classes. In the revision, we clarified this explicitly: the example $f(t)=t^2$ certifies the sharp first-order behavior for the non-bilinear methods, whereas a higher-degree polynomial is needed to expose the sharp second-order behavior of bilinear, which is why we use $f(t)=t^3$ in that case. We revised the remark accordingly to avoid the impression of an asymmetric comparison.
>
> [1] A. Gu, K. Goel, and C. Ré. Efficiently modeling long sequences with structured state spaces. International Conference on Learning Representations, 2022.

---

### Review · Reviewer_ePBo · 2025-12-19

**Summary Of Contributions:**

The paper provides rigorous theoretical foundations for the HiPPO-LegS ODE used in deep state-space models. It proves the ODE is well-posed despite its singularity, establishes convergence of standard numerical discretization schemes for Riemann-integrable inputs.

**Audience:**

Yes

**Audience Explanation:**

The HiPPO framework has seen widespread adoption (e.g., in models such as S4, S5, and Mamba), yet the discretization methods used in practice have often lacked theoretical justification. This paper fills a critical gap by offering the first rigorous convergence analysis of these discretization schemes under realistic conditions, including discontinuous or non-smooth input signals. As a result, the findings are highly relevant to TMLR’s audience, particularly researchers working on continuous-time models, recurrent architectures, state space models (SSMs), and numerical methods in machine learning.

**Broader Impact Concerns:**

The paper is theoretical in nature and does not introduce new models or datasets. It provides foundational analysis that may lead to more stable and reliable implementations of state-space models, but does not raise ethical, societal, or security concerns. The work is squarely in the domain of mathematical machine learning and numerical analysis.

**Claims And Evidence:**

Yes

**Claims Explanation:**

1.	The theoretical claims are substantiated through careful mathematical derivations, lemmas, and theorems.
2.	Numerical experiments have been conducted to validate the theoretical rates across different input regularities

**Requested Changes:**

1.	It is recommended to add a summary table summarizing methods, assumptions on f, and convergence rates. It would provide a quick reference and improve readability.
2.	Some notations appear abruptly without prior and clear explanation, such as “right-hand-side g” in Page 4, and Θ(1/n) and Θ(1/n2) in Page 16.
3.	The ending of the article feels somewhat abrupt. It is recommended to add a dedicated "Conclusion" section to systematically summarize the key contributions and provide a brief outlook on future research directions. This would enhance the overall structure and impact of the paper.
4.	There are some typos and grammatical errors:
Page 2: “The SSM architecture has been…, and are used across” -> “is used across”;
Page 2: “While LMU and its variants has proven” -> “have proven”

---

> ### Author Response · Authors · 2026-04-21
>
> We thank the reviewer for the positive assessment and helpful suggestions on presentation.
> We agree with these points and revised the paper accordingly. In particular, we added a summary table in the introduction collecting the discretization methods, assumptions on $f$, and the convergence guarantees proved in the paper. We also introduced notation more carefully where it first appears, including the asymptotic notation used in the convergence-rate discussion. In addition, we added a brief conclusion section summarizing the main contributions and natural future directions. Finally, we corrected the typos and grammatical issues pointed out by the reviewer.

---

### Review · Reviewer_iKiH · 2026-03-11

**Summary Of Contributions:**

This paper studies the mathematical foundations of the HiPPO-LegS (Lagendre Scaled) ODE, a singular state-space ODE used in deep state space models to encode the history of an input function $f$ through Legendre projections. Despite the practical relevance of HiPPO-based memory units in modern SSMs, the underlying singular ODE and the discretization schemes used in existing work have not been investigated from a mathematical perspective, which is studied in this paper. Specifically, this paper has shown the following results: (1) The HiPPO-LegS ODE is well posed (2) Several commonly-used discretization schemes converge under weak assumptions and their associated  convergence rates (3) Numerical verifications of the claims above.

**Additional Comments:**

Overall, the reviewer thinks that this paper studied an important topic, but it needs some refinement (especially a larger set of empirical validations) before being considered for TMLR.

References:

[1] Gu, Albert, Tri Dao, Stefano Ermon, Atri Rudra, and Christopher Ré. "Hippo: Recurrent memory with optimal polynomial projections." Advances in neural information processing systems 33 (2020): 1474-1487.

**Audience:**

No

**Audience Explanation:**

My answer to question above will be somewhat a mixture of "Yes" and "No", which I explain in detail below.

State space models (with HiPPO-style memory mechanisms) and their theoretical analyses remain an active area in current deep learning research. To the best of the reviewer’s knowledge, this paper addresses a foundational issue that is missing in the literature: whether the singular HiPPO-LegS ODE is well posed and whether the discretizations used in practice are provably convergent under structural regularity assumptions on the input. Hence, I do think a few TMLR audience, especially those who work on machine learning theory, will find the results valuable and interesting.

However, the reviewer also thinks it will be important to expand the empirical validation part to increase the exposure of the paper (for instance, to practitioners who focus on the empirical aspects of SSM). Please refer to the "Requested Changes" section for specific empirical validations to be added.

**Claims And Evidence:**

Yes

**Claims Explanation:**

The paper’s main claims are mostly theoretically mathematical, whose detailed proofs have been provided.

**Requested Changes:**

The reviewer thinks that the current numerical section is probably a bit minimal. A somewhat stronger validation would help - the authors are strongly encouraged to perform similar experiments as that of [1] to justify their theoretical findings. Some other experiments that the authors might consider adding include:

(1) An experiment illustrating the practical difference between bilinear and approximate bilinear schemes in a toy model of sequence-modeling task,

(2) A numerical comparison (similar to ablation studies) of constants across the schemes (not just asymptotic slopes).

---

> ### Author Response · Authors · 2026-04-21
>
> We thank the reviewer for recognizing the importance of establishing rigorous foundations for the HiPPO-LegS ODE.
>
> Regarding the reviewer's first suggestion, we respectfully argue that such an end-to-end experiment may not be the most direct way to validate the core mathematical claims of this paper, for two reasons. First, our contribution is a numerical analysis of the HiPPO-LegS ODE itself, and thus the most faithful empirical validation is to directly measure discretization errors, convergence rates, and finite-n behavior, rather than downstream performance in a learned sequence-modeling pipeline, which would introduce confounding effects from architecture, parametrization, and optimization.
> Second, our theory already explains why a clear practical gap between the bilinear and approximate bilinear schemes need not appear in such a toy task. In the nonsmooth regime covered by our theory, both schemes have the same convergence order; bilinear gains a higher-order advantage only under additional smoothness assumptions on the input. Since many practically relevant sequence-modeling inputs do not naturally satisfy such extra smoothness assumptions, one should not expect a toy sequence-modeling experiment to provide a clean or systematic separation between these two schemes.
> From this perspective, our work could be seen as justifying the use of approximate bilinear method for sequence-modeling tasks.
>
> Regarding the reviewer's second point, we agree that comparing only asymptotic slopes is incomplete.
> In the revision, we therefore added a numerical comparison of finite-n error constants across schemes in addition to the original convergence plots. Together, these additions make the practical differences between discretization choices more transparent. Please refer to the General Response for details.

---

### Review · Reviewer_nD72 · 2026-03-25

**Summary Of Contributions:**

This paper analyzes the singular HiPPO-LegS ODE used in deep state-space models. It proves well-posed problem under restricted initial conditions, interprets discretizations as quadrature rules by establishing a convergence for Riemann integrable inputs. A derivation of convergence rates under additional regularity assumptions have been given with theoretical justification for discretization schemes used in modern sequence modeling schemes.

**Additional Comments:**

None

**Audience:**

Yes

**Audience Explanation:**

The researchers working on state-space models, sequence modeling, HiPPO, and theoretical foundations of deep learning architectures would be interested in these findings. In general, it can be said that the people who likes theoretical research would be interested.

**Claims And Evidence:**

Yes

**Claims Explanation:**

The claims are mathematically motivated and supported, but several arguments requires rigor, clarity, and completeness, that will strengthen the overall evidence.

**Requested Changes:**

The paper addresses an important theoretical gap in HiPPO-LegS state-space models. However, several proofs require stronger justification, and the improvements in stability analysis and empirical validation can improve the quality of presentation.
The authors are suggested to explore following points:

1. Strengthen mathematical rigor of key proofs: Clarify assumptions and improve justification in the well-posedness and convergence analysis, particularly in Lemma 5 and Theorem 2, where convergence of the numerical solution ||c_n - c(t_n)|| tends to 0 is established.

2. Add stability analysis of the continuous and discrete systems: In addition to convergence, analyze boundedness and numerical stability of the singular system defined by c'(t) in eqn 1, and discuss long-time behavior and sensitivity to perturbations.

3. The requirement c(0)=f(0)e_1, is strong and may not hold in practical implementations. The authors should discuss its implications and whether approximate or relaxed solutions are possible.

4. Include experiments comparing discretization schemes and empirically verifying convergence behavior and theoretical rates.

5. Simplify proofs, add intuition behind key lemmas, and improve notation consistency to enhance readability.

6. Discuss implications for modern state-space models (e.g., S4, Mamba), particularly how the theoretical findings influence discretization choices and implementation.

7. Expand related work and positioning: Better situate the work within neural ODEs, state-space models, and singular ODE literature.

---

> ### Author Response · Authors · 2026-04-21
>
> We thank the reviewer for the careful reading and thoughtful suggestions. In the revision, we strengthened the presentation of the key mathematical arguments, especially around Corollary 2 and Theorem 2, by correcting the missing endpoint assumption, clarifying the role of the fixed-degree interpolants, and expanding the justification of the convergence argument in Theorem 2.
>
> Regarding stability, we agree that some discussion of boundedness is useful, and we added a brief finite-horizon stability argument in Remark 3.3. Specifically, the boundedness and smooth dependence on input follow from Corollary 1. We do not think that full long-time / infinite horizon analysis is necessary for the current paper, since the finite horizon is exactly the setting used by the HiPPO memory operator.
>
> Regarding the condition $c(0)=f(0)e_1$, we would like to emphasize that this is not an extra modeling assumption we impose; it is a structural consequence of the exact singular ODE if one asks for a continuous solution on $[0,T]$. Implementations that use a different initialization, or avoid the singular step via an index shift should be interpreted as approximating a different dynamics rather than the exact LegS ODE.

---

### Author Response · Authors · 2026-04-21

# General Response

Dear Action Editor and Reviewers,

We thank all reviewers for their careful reading and constructive feedback. We were encouraged that the reviewers found the paper technically sound and recognized that it addresses an important theoretical gap in the foundations of HiPPO-LegS. The main suggestions were to improve clarity, make several proof steps more explicit, and strengthen the numerical experiments section. We agreed with these points and revised the paper accordingly.

First, we improved readability and presentation throughout. In particular, we revised the introduction around Eq (1) to explain the role of $c(t)$ more clearly. In addition, we added a summary table in the introduction collecting the discretization schemes, assumptions on $f$, and corresponding convergence guarantees, and we added a short conclusion section summarizing the main contributions and natural future directions.

Second, we strengthened the numerical section. In addition to the original convergence plots, we added a table comparing finite-$n$ error constants across discretization schemes, with the formula
$$ E_n \approx C n^{-p} $$
 where $E_n$ is the discretization error with $n$ meshgrid points, $p$ is the order of accuracy, and $C$ is the associated constant.
This makes the practical differences between methods with the same asymptotic order more transparent.


Third, we clarified several proof steps, especially in Corollary 2 and Theorem 2.
For Corollary 2, we added to the corollary statement the assumption of the uniform boundedness of the endpoint coefficients. We also made it clear where this assumption is verified in Theorem 2.
Moreover, in the proof of Theorem 2, we added details to the part where the pointwise convergence of the interpolant function is strengthened to uniform convergence, making it explicit where the degree boundedness of the interpolant polynomials are used.

---

### Comment · Action_Editor_QMqY · 2026-06-24
**Typos to fix**

Dear authors,

Thanks for putting up the camera ready; and for including all the necessary content discussed with the reviewers.

I spotted a few typos which could enhance the paper:

  - p.2: "the transformers architecture" → "the transformer architecture"
  - p.3: "constructing a $N$-dimensional ODE" → "an $N$-dimensional ODE"
  - p.5: "$\tilde c_j$ is $j$-th component function" → "is the $j$-th"
  - p.6: "is satisfied if and only the existence" → "if and only if"
  - p.6: "By fundamental theorem of calculus" → "By the fundamental…" (also again in Theorem 1 proof)
  - p.6: "Since $\hat c_j$ is a solution it is continuous" → add comma after "solution"
  - p.7: "We prove by showing" → "We prove this by showing"
  - p.12: "triangular inequality" → "triangle inequality" (used correctly elsewhere)
  - p.13: "is a $i-1$ degree polynomial" → "is an $(i-1)$-degree polynomial" (article + parens), same latter with j
  - p.13: "is at most $j-1$ degree polynomial,i.e.," → "an at-most-$(j-1)$-degree polynomial, i.e.," (missing space + article)
  - p.17, Remark 5.2: "(The bilinear method yields the exact solution in this case)" → "in this case.)" (missing period before paren)
  - App. A: "it is suffices to show" → "it suffices to show"
  - App. B: "regarding as $x$ as a symbolic" → "regarding $x$ as" (delete duplicate "as")
  - App. D: "converges to equation 13 pointwise as $n\to\infty$ for $x\in[0,1]$ as $n\to\infty$": duplicate trailing "as $n\to\infty$"; remove one. Also "is a $j-1$ degree polynomial" same article/paren issue as
- Capitalization nitpick: sometimes you use "mamba" (lowercase) but the rest of the paper uses "Mamba".

Once these are fixed I will accept the camera ready version !

Thanks,
AE

---

> ### Author Response · Authors · 2026-06-25
>
> Dear AE,
>
> We sincerely appreciate your time and careful review. We have uploaded the revised camera-ready PDF (camera-ready-revision) and addressed all of the typographical corrections you listed.
>
> Thank you very much.
>
> Best regards,
> The authors

---

### Decision · Action_Editor_QMqY · 2026-05-26

**Recommendation:** Accept as is

**Audience:**

Yes

**Audience Explanation:**

The HiPPO framework fits many widely-used models (S4, Mamba), yet the mathematical well-posedness of the LegS ODE and the convergence guarantees of its discretizations had not been established before. This paper fills that gap

Researchers working on state space models and their theoretical foundations will find these results valuable.

As Reviewer_ePBo noted: "the discretization methods used in practice have often lacked theoretical justification. This paper fills a critical gap by offering the first rigorous convergence analysis of these discretization schemes under realistic conditions". Reviewer_iKiH similarly acknowledged that "this paper addresses a foundational issue that is missing in the literature".

**Claims And Evidence:**

Yes

**Claims Explanation:**

The claims are purely mathematical and are supported by detailed, rigorous proofs. The key insight about reinterpreting discretization schemes as quadrature rules on the input function is elegant and enables a convergence proof that bypasses the failure of standard LTE-based analysis due to the singularity at t=0.

Numerical experiments cleanly validate the theoretical rates across smooth, bounded-variation, and Riemann-integrable inputs. The revised manuscript strengthens the evidence with finite error constant comparisons and clarified proof steps (Corollary 2, Theorem 2).

All four reviewers agreed that the mathematical claims are supported (Reviewer_AfdB: "I have no doubt the mathematical part is rigorous"; Reviewer_ePBo: "theoretical claims are substantiated through careful mathematical derivations"; Reviewer_iKiH: "detailed proofs have been provided"; Reviewer_nD72: "claims are mathematically motivated and supported").